# A Generalization Bound for Nearly-Linear Networks

**Eugene Golikov**                                           *evgenii.golikov@epfl.ch*
*Chair of Statistical Field Theory*
*École Polytechnique Fédérale de Lausanne (EPFL)*

**Reviewed on OpenReview:** *https://openreview.net/forum?id=tRpWaK3pWh*

## Abstract

We consider nonlinear networks as perturbations of linear ones. Based on this approach, we present a novel generalization bound that become non-vacuous for networks that are close to being linear. The main advantage over the previous works which propose non-vacuous generalization bounds is that our bound is *a priori*: performing the actual training is not required for evaluating the bound. To the best of our knowledge, it is the first non-vacuous generalization bound for neural nets possessing this property.

## 1 Introduction

Despite huge practical advancements of deep learning, the main object of this field, a neural network, is not yet fully understood. As we do not have a complete understanding of how neural networks learn, we are not able to answer the main question of deep learning theory: *why do neural networks generalize on unseen data?*

While the above question is valid for any supervised learning model, it is notoriously difficult to answer for neural nets. The reason is that modern neural nets have billions of parameters and as a result, huge capacity. Therefore among all parameter configurations that fit the training data, there provably exist such configurations that do not fit the held-out data well (Zhang et al., 2021). This is the reason why classical approaches for bounding a generalization gap, i.e. the difference between distribution and train errors, fall short on neural networks: such approaches bound the gap uniformly over a model class. That is, if weights for which the network performs poorly exist, we bound our trained network's performance by performance of that poor one.

As we observe empirically, networks commonly used in practice do generalize, which means that training algorithms we use (i.e. gradient descent or its variants) choose "good" parameter configurations despite the existence of poor ones. In other words, these algorithms are *implicitly biased* towards good solutions.

Unfortunately, implicit bias of gradient descent is not fully understood yet. This is because the training dynamics is very hard to integrate, or even characterize, analytically. There are two main obstacles we could identify. First, modern neural networks have many layers, resulting in a high-order weight evolution equation. Second, activation functions we use are applied to hidden representations elementwise, destroying a nice algebraic structure of stacked linear transformations.

If we remove all activation functions, the training dynamics of gradient descent can be integrated analytically under certain assumptions (Saxe et al., 2013). However, the resulting model, a linear network, is as expressive as a linear model, thus loosing one of the crucial advantages of neural nets.

**Idea.** The idea we explore in the present paper is to consider nonlinear nets as perturbations of linear ones. We show that the original network can be approximated with a proxy-model whose parameters can be computed using parameters of a linear network trained the same way as the original one. Since the proxy-model uses the corresponding linear net's parameters, its generalization gap can be meaningfully bounded with classical approaches. Indeed, if the initial weights are fixed, the result of learning a linear network to minimize square loss on a dataset $(X \in \mathbb{R}^{d \times m}, Y \in \mathbb{R}^{d_{out} \times m})$ is determined uniquely by $YX^{\top} \in \mathbb{R}^{d_{out} \times d}$ and

$XX^\top \in \mathbb{R}^{d \times d}$, where $d, d_{out}$ are the input and output dimensions, and $m$ is the dataset size. The number of parameters in these two matrices is much less than the total number of parameters in the network, making classical counting-based approaches meaningful.

**Contributions.** Our main contribution is a generalization bound given by Theorem 5.2, which is ready to apply in the following setting: (1) fully-connected networks, (2) gradient descent with vanishing learning rate (gradient flow), (3) binary classification with MSE loss. The main disadvantage of our bound is that it diverges as training time $t$ goes to infinity. We discuss how to choose the training time in such a way that the bound stays not too large while the training risk diminishes significantly, in Section 5.3. We validate our bound on a simple fully-connected network trained on a downsampled MNIST dataset, and demonstrate that it becomes non-vacuous in this scenario (Section 6). We list advantages and disadvantages of our bound over that of existing generalization bounds in Section 2. We discuss the assumptions we use, as well as possible improvements of our bound, in Appendix A. Finally, we discuss how far the approach we choose in this work, i.e. generalization bounds based on deviation from specific proxy models, could lead us in the best case scenario (Appendix B).

## 2 Comparison to previous work

**Advantages.** The bound of our Theorem 5.2 has the following advantages over some other non-vacuous bounds available in the literature (e.g. Biggs & Guedj (2022); Galanti et al. (2023); Dziugaite & Roy (2017); Zhou et al. (2019)):

1. It is an *a priori* bound, i.e. getting the actual trained the model is not required for evaluating it. To the best of our knowledge, it is the first non-vacuous *a priori* bound available for neural nets. All works mentioned above, despite providing non-vacuous bounds, could be evaluated only on a trained network thus relying on the implicit bias phenomenon which is not well understood yet.

2. Related to the previous point, to the best of our knowledge, our bound is the first to incorporate the implicit bias of gradient flow explicitly. This is done by demonstrating that the model does not deviate much from a specific proxy model. The class of realizable proxy models has a small number of effective parameters. Informally, this indicates that this class is "simple"; hence the gradient flow is biased towards this simple class, and our bound exploits this property.

3. It does not require a held-out dataset to evaluate it, compared to Galanti et al. (2023) and coupled bounds of Biggs & Guedj (2022). Indeed, if one had a held-out dataset, they could just use it directly to evaluate the trained model, thus questioning the practical utility of such bounds.

4. It does not grow with network width. In contrast, PAC-Bayesian bounds, Biggs & Guedj (2022); Dziugaite & Roy (2017); Zhou et al. (2019), might grow with width.

5. Similarly to Biggs & Guedj (2022); Galanti et al. (2023), we bound the generalization gap of the original trained model, not its proxy. In contrast, Dziugaite & Roy (2017) introduces a Gaussian noise to the learned weights, while Zhou et al. (2019) crucially relies on quantization and compression after training.

Related to the second point, we bound the generalization gap for the proxy model with a simple parameter-counting bound (similar to that of Vapnik & Chervonenkis (1971)), thus demonstrating that, contrary to a common judgement, such bounds could be useful in the context of models with many more parameters than data points.

**Disadvantages.** For a fair comparison, we also list the disadvantages our present bound has:

1. The bound of our Theorem 5.2 becomes non-vacuous only when the following two conditions hold. First, a simple counting-based generalization bound for a linear model evaluated in the same setting should be non-vacuous. Such a bound is vacuous even for binary classification on the standard MNIST dataset, but becomes non-vacuous if we downsample the images.

2. Second, the activation function has to be sufficiently close to being linear. To be specific, for a two-layered leaky ReLU neural net trained on MNIST downsampled to 7x7, one needs the negative slope to be not less than 0.99 (1 corresponds to a linear net, while ReLU corresponds to 0).

3. One may hope for the bound to be non-vacuous only for a partially-trained network, while for a fully-trained network the bound diverges.

4. Even in the most optimistic scenario, when we manage to tighten the terms of our bound as much as possible, our bound stays non-vacuous only during the early stage of training when the network has not started "exploiting" its nonlinearity yet (Appendix B). However, the minimal negative ReLU slope for which the bound stays non-vacuous is much smaller, 0.6.

## 3 Related work

**Non-vacuous generalization bounds.** While first generalization bounds date back to Vapnik & Chervonenkis (1971), the bounds which are non-vacuous for realistically large neural nets appeared quite recently. Specifically, Dziugaite & Roy (2017) constructed a PAC-Bayesian bound which was non-vacuous for small fully-connected networks trained on MNIST and FashionMNIST. The bound of Zhou et al. (2019), also of PAC-Bayesian nature, relies on quantization and compression after training. It ends up being non-vacuous for VGG-like nets trained on large datasets of ImageNet scale. The bound of Biggs & Guedj (2022) is the first non-vacuous bound that applies directly to the learned model and not to its proxy. It is not obvious whether their construction could be generalized to neural nets with more than two layers. The bound of Galanti et al. (2023) is not PAC-Bayesian in contrast to the previous three. It is based on the notion of effective depth: it assumes that a properly trained network has small effective depth, even if it is deep. Therefore its effective capacity is smaller than its total capacity, which allows for a non-vacuous bound. See the previous section for discussion of some of the features of these bounds.

**A priori and a posteriori generalization bounds.** Most of the generalization bounds available in the literature are *a posteriori*. That is, some PAC-Bayesian bounds (Dziugaite & Roy, 2017; Neyshabur et al., 2018; Biggs & Guedj, 2022), as well as Rademacher complexity-based bounds (Bartlett et al., 2017), depend on norms of learned weights, or distances between the learned weights and their initializations, so they cannot be computed before these norms or distances are known. The bound of Zhou et al. (2019) depends on the trained model size after compression and quantization, which cannot be predicted before training and compression. That is, this bound implicitly relies on the fact that the training procedure prefers well-compressible models; to the best of our knowledge, this fact does not have a solid explanation yet. The bound of Galanti et al. (2023) depends on a so-called *effective depth* of a learned model, which also cannot be predicted in advance.

The only *a priori* bounds we are aware of are the classical uniform bounds based on VC-dimension and Rademacher complexity, see Vapnik & Chervonenkis (1971). These bounds grow with model expressivity, hence with both width and depth, which make them vacuous in most of the practical scenarios involving neural nets. The reason they are vacuous is also related to the fact that among all models that interpolate the data, one can often find a model that does not generalize well, see empirical results of Zhang et al. (2021). Uniform bounds bound the generalization gap in the worst case, hence if a "bad" model exists in our model class, the whole generalization bound has no chance to be good.

Overall, generalization bounds that use some complexity of the whole function class fall short due to the trade-off they impose: low complexity = provably good generalization, high complexity = maybe poor generalization. Modern neural architectures are deep and wide, hence they have high complexity, while they still generalize well. The reason is that the training procedure we often use is likely to be implicitly biased towards well-generalizing solutions. If we bound the generalization gap uniformly over the whole function class, we ignore this effect. The *a posteriori* bounds we discuss above do take some form of implicit bias into account (i.e. low weight norm, compressibility etc.). So does our bound: we exploit the fact that a nearly linear network is close to a proxy which uses only weights of a linear network. In their turn, the trained weights of a linear network depend only on a few parameters as long as their initialization is given. The

difference is that our bound does not need any prior info about the learned weights (only on their initialization and the optimization procedure); hence it is *a priori.*

**Linear networks training dynamics.** The pioneering work that integrates the training dynamics of a linear network under gradient flow to optimize square loss is Saxe et al. (2013). This work crucially assumes that the initial weights are well aligned with the eigenvectors of the optimal linear regression weights $YX^+$, where $(X, Y)$ is the train dataset. As the initialization norm approaches zero, the learning process becomes more sequential: components of the data are learned one by one starting from the strongest. Li et al. (2021) conjecture that the same happens for any initialization approaching zero excluding some set of directions of measure zero. They prove this result for the first, the strongest, component, but moving further seems more challenging. See also Yun et al. (2021); Jacot et al. (2021). We note a recent result of Tu et al. (2024) who propose a so-called *mixed* dynamics for two-layered linear networks that does not suffer from this issue.

**Nearly-linear neural nets.** Our generalization gap bound decreases as activation functions get closer to linearity. While nearly-linear activations do not conform with the usual practice, nearly-linear networks have been studied before. That is, Li et al. (2022) demonstrated that when width $n$ and depth $L$ go to infinity with constant ratio, the hidden layer covariances at initialization admit a meaningful limit as long as the ReLU slopes behave as $1 \pm \frac{c}{\sqrt{n}}$, i.e. become closer and closer to linearity. Noci et al. (2024) explored a similar limit for Transformers (Vaswani et al., 2017). Another example is Kumar et al. (2024), who used a toy small-$\epsilon$ model to explain the phenomenon of *grokking* (Power et al., 2022) as delayed feature learning.

**Linearized training of neural networks and NTK.** It is important to emphasize that linear networks we consider in the present work are fundamentally different from the NTK model introduced by Jacot et al. (2018) resulted from linearized training of a neural network. That is, by a linear network we mean a network $f_\theta(x)$ who is linear in its input, $x$, but not necessarily in its weights $\theta$. Whereas if we linearize the training procedure, the trained model becomes linear in $\theta$, but not in $x$. Jacot et al. (2018) demonstrated that training a neural network becomes equivalent to training a kernel method under specific (non-standard) parameterization as width goes to infinity (NTK limit). Since training a kernel method is equivalent to training a random feature model with sufficiently large number of features, the corresponding training procedure is indeed linear in weights. See also Yang & Littwin (2021) for proving NTK limit for general architectures, and Chizat et al. (2019) for demonstrating linearized training for large weight initializations.

As in the NTK limit training a neural net becomes equivalent to training a kernel method, features the model uses stay the same over the whole process of training. In other words, the model does not enjoy *feature learning* when training is linearized. In contrast, even a linear network with two layers trained with gradient descent does demonstrate feature learning: see Saxe et al. (2013); Tu et al. (2024). The feature learning it exhibits is kernel alignment and a so-called *momentum effect*: the associated kernel (NTK), or equivalently, the features the model uses, aligns over strong components of the data and extends along them Tu et al. (2024). This way, these strong components get learned first, before weak components, often resulted from data noise.

We also emphasize that the proxy models we use to prove our results, while exploiting weights of a trained linear network, are not linear themselves; neither in $x$, nor in weights. Therefore they do not suffer from expressivity issues as linear networks (which are all functionally equivalent to linear models $x \to Wx$), while they do enjoy feature learning (since linear networks do).

Therefore our approach is quite orthogonal to NTK and generalizability of kernel methods. In terms of high-level methodology, the closest paper we could mention is Arora et al. (2019), where they consider a two-layer NTK-parameterized MLP, and prove a generalization bound for it when width is sufficiently large. For this, they combine a generalization bound for the infinite-width limit when the kernel is constant, and a term that takes into account the fact that the kernel for finite width deviates from the limit one. Whereas what we do is we combine a generalization bound for a proxy model that exploits the weights of a linear network and a proxy deviation bound. The bound of Arora et al. (2019) becomes good when the width is large (hence the finite-width NTK is close to the limit NTK), while our bound becomes good when the activation functions are close to be linear (hence the proxy does not deviate from the original model much).

## 4    Our approach

Before formally stating our main bound, we present the high-level approach for constructing generalization bounds we take in our work.

**Generalization bound based on proxy models.**    Let $f$ be a model trained on a binary classification task. We will look for a proxy model $g$ that satisfies the following properties: (1) it is close enough to $f$, and (2) the generalization gap for $g$ could be bounded well enough. Given such $g$, we bound the distribution risk of $f$ as follows:

$$R[f] \leq \hat{R}_\gamma[f] + \left(R^C_\gamma[g] - \hat{R}^C_\gamma[g]\right) + \frac{1}{\gamma}\mathbb{E}_{x\sim\mathcal{D}}|f(x) - g(x)| + \frac{1}{\gamma}\frac{1}{m}\sum_{k=1}^m|f(x_k) - g(x_k)|, \tag{1}$$

where $\gamma > 0$ and we used three different risk functions:

1. Misclassification risk: $r(y, \hat{y}) = [y\hat{y} < 0]$,

2. Margin risk: $r_\gamma(y, \hat{y}) = [y\hat{y} < \gamma]$,

3. Continuous margin risk: $r^C_\gamma(y, \hat{y}) = [y\hat{y} < 0] + \left(1 - \frac{y\hat{y}}{\gamma}\right)[y\hat{y} \in [0, \gamma]]$,

giving rise to the distribution risk $R[f] = \mathbb{E}_{x,y\sim\mathcal{D}}r(y, f(x))$, and similarly $R_\gamma[f]$ and $R^C_\gamma[f]$, and its empirical counterpart $\hat{R}[f] = \frac{1}{m}\sum_{k=1}^m r(y_k, f(x_k))$, and similarly $\hat{R}_\gamma[f]$ and $\hat{R}^C_\gamma[f]$. Here $\mathcal{D}$ is the data distribution, and $(x_k, y_k)_{k=1}^m$ is the training dataset samled from $\mathcal{D}$ in an iid manner.

We derived Equation (1) simply from the fact that $r^C_\gamma(\cdot, y)$ is $1/\gamma$-Lipschitz, and $r \leq r^C_\gamma \leq r_\gamma$, see Equation (22) below for a more elaborate derivation. The first term on the right hand side of Equation (1) is the empirical margin risk of the original model $f$. The second term is the generalization gap of the proxy-model $g$ (with respect to continuous margin risk) which we assume to be easy to bound. The two remaining terms are deviations of $g$ from $f$ averaged over the whole data distribution $\mathcal{D}$ and the dataset, divided by $\gamma$. The parameter $\gamma$ controls the trade-off between the first term and the other three: the first term increases with $\gamma$ (eventually reaching 1), while the other three decrease.

**Proxy models for a nearly-linear network.**    The problem now boils down to finding a good proxy $g$ that approximates the trained model $f$ well, and whose generalization gap could be well-bounded. Let $f^\epsilon_\theta(x)$ be a neural network with parameters $\theta$, input $x \in \mathbb{R}^d$, scalar output, and activation functions $\phi^\epsilon(z) = z + \epsilon\psi(z)$, $\psi$ is a positively homogeneous map (e.g. ReLU). That is, $f^0_\theta(x)$ is linear in $x$ (but not necessarily in $\theta$), and $\epsilon$ characterizes the deviation from linearity.

Let $f^\epsilon_{\theta^\epsilon}$ be the trained model whose generalization gap we would like to bound, and let $f^0_{\theta^0}$ be a linear model trained the same way. We consider the following proxies: $g_1(x) = f^\epsilon_{\theta^0}(x)$ and $g_2(x) = g_1(x) + (f^\epsilon_{\theta^\epsilon}(X) - g_1(X))X^+x$, where $X \in \mathbb{R}^{d\times m}$ is a matrix with rows $(x_k)_{k=1}^m$, and $X^+$ is its Moore-Penrose pseudo-inverse.

In words, the first proxy is a nonlinear network that uses weights of the trained linear one, while the second proxy results from linearly correcting the first one on the train dataset. These proxies deviate from the original model as follows: $|g_\kappa(x) - f^\epsilon_{\theta^\epsilon}(x)| = O(\epsilon^\kappa)$ for $\kappa \in \{1, 2\}$ and any given $x \in \mathbb{R}^d$.

Under certain assumptions, $\theta^0$, the trained linear model weights, depend only on $YX^+ \in \mathbb{R}^d$, where $Y \in \mathbb{R}^{1\times d}$ is a row-vector of targets. Since $g_1$ is fully described by $\theta^0$, the class of models realizable by it has only $d$ parameters. Since $g_2$ is merely $g_1$ plus a linear correction, its respective class of models has only $d + d = 2d$ parameters. In both cases, the number of parameters the proxy has is much smaller than the total number of weights of the network, which is $(d + 1)n + (L - 2)n^2$. Then classical uniform bound techniques (Vapnik & Chervonenkis, 1971) result in a generalization bound for proxy models that grows as $\sqrt{\frac{\kappa d}{m}}$. This bound depends neither on width of the network, nor on its depth.

# 5 Main result

**Notations.** For integer $L \geq 0$, $[L]$ denotes the set $\{1, \ldots, L\}$. For integers $l \leq L$, $[l, L]$ denotes the set $\{l, \ldots, L\}$. For a vector $x$, $\|x\|$ denotes its Euclidean norm, while $\|x\|_1$ denotes its $l_1$-norm. For a matrix $X$, $\|X\|$ denotes its maximal singular value, while $\|X\|_F$ denotes its Frobenius norm.

## 5.1 Setup

**Model.** The model we study is a fully-connected LeakyReLU network with $L$ layers:

$$f_\theta^\epsilon(x) = W_L x_{\theta,L-1}^\epsilon(x), \qquad x_{\theta,0}^\epsilon(x) = x, \qquad x_{\theta,l}^\epsilon(x) = \phi^\epsilon \left( W_l x_{\theta,l-1}^\epsilon(x) \right) \quad \forall l \in [L-1], \tag{2}$$

where $\theta \in \mathbb{R}^N$ denotes the vector of all weights, i.e. $\theta = \text{cat}(\{\text{vec}(W_l)\}_{l=1}^L)$, $W_l \in \mathbb{R}^{n_l \times n_{l-1}} \ \forall l \in [L]$, and $\phi^\epsilon$ is a Leaky ReLU with a negative slope $1 - \epsilon$, that is:

$$\phi^\epsilon(x) = x - \epsilon \min(0, x). \tag{3}$$

Since we are going to consider only binary classification in the present work, we take $n_L = 1$. We also define $d = n_0$ to denote the input dimension.

**Data.** Data points $(x, y)$ come from a distribution $\mathcal{D}$. We assume all $x$ from $\mathcal{D}$ to lie on a unit ball, $\|x\| \leq 1$, and $y \in \{-1, 1\}$. During the training phase, we sample a set of $m$ points iid from $\mathcal{D}$ to form a dataset $(X, Y)$, where $X \in \mathbb{R}^{d \times m}$ and $Y \in \mathbb{R}^{1 \times m}$.

**Training.** Assuming $\text{rk}\, X = d$ (which implies $m \geq d$, i.e. the data is abundant), we train our model on $(X, Y)$ with gradient flow to optimize square loss on whitened data, i.e. on $(\tilde{X}, Y)$ for $\tilde{X} = \Sigma_X^{-1/2} X$, where $\Sigma_X = \frac{1}{m} X X^\top \in \mathbb{R}^{d \times d}$ is an empirical feature correlation matrix. That is,

$$\frac{dW_l^\epsilon(t)}{dt} = -\frac{\partial \left( \frac{1}{2m} \left\| Y - f_{\theta^\epsilon(t)}^\epsilon(\tilde{X}) \right\|_F^2 \right)}{\partial W_l} \quad \forall l \in [L]. \tag{4}$$

Note that $\tilde{X} \tilde{X}^\top = m I_d$.

**Inference.** To conform with the above training procedure, we take the model output at a point $x$ to be $f_\theta^\epsilon \left( \Sigma^{-1/2} x \right)$, where $\Sigma$ is a (distribution) feature correlation matrix: $\Sigma = \mathbb{E}_{(x,y) \sim \mathcal{D}}(x x^\top) \in \mathbb{R}^{d \times d}$. We assume this matrix to be known; in practice, we could substitute it with $\Sigma_X$.

**Performance measure.** Recall the definitions of $r$, $r_\gamma$, and $r_\gamma^C$ from Section 4. Slightly abusing the notation, we define the empirical (train) risk on the dataset $(X, Y)$ and the distribution risk of the model trained for time $t$ as

$$\hat{R}^\epsilon(t) = \frac{1}{m} \sum_{k=1}^m r\left( y_k, f_{\theta^\epsilon(t)}^\epsilon(\Sigma^{-1/2} x_k) \right), \qquad R^\epsilon(t) = \mathbb{E}_{(x,y) \sim \mathcal{D}}\, r\left( y, f_{\theta^\epsilon(t)}^\epsilon(\Sigma^{-1/2} x) \right). \tag{5}$$

We define $R_\gamma^\epsilon(t)$ and $R_\gamma^{C,\epsilon}(t)$ analogously to $R^\epsilon(t)$, and $\hat{R}_\gamma^\epsilon(t)$ and $\hat{R}_\gamma^{C,\epsilon}(t)$ analogously to $\hat{R}^\epsilon(t)$.

## 5.2 Generalization bound

We will need the following assumption on the training process:

**Assumption 5.1.** $\forall t \geq 0 \ \left\| \left( Y - f_{\theta^\epsilon(t)}^\epsilon(\tilde{X}) \right) \tilde{X}^\top \right\|_F^2 \leq \left\| \left( Y - f_{\theta^\epsilon(0)}^\epsilon(\tilde{X}) \right) \tilde{X}^\top \right\|_F^2.$

Note that $\left\| Y - f_{\theta^\epsilon(t)}^\epsilon(\tilde{X}) \right\|_F^2 \leq \left\| Y - f_{\theta^\epsilon(0)}^\epsilon(\tilde{X}) \right\|_F^2$ since we minimize the loss monotonically with gradient flow. This implies that the above assumption holds automatically, whenever $\tilde{X}$ is a (scaled) orthogonal matrix

(which happens when $m = d$). It is easy to show that it provably holds for a linear network ($\epsilon = 0$), see Appendix E, and we found this assumption to hold empirically for all of our experiments with nonlinear networks too, see Figure 7.

We are now ready to formulate our main result:

**Theorem 5.2.** *Fix $\beta, \gamma > 0$, $t \geq 0$, $\delta \in (0,1)$, $\epsilon \in [0,1]$, and $\kappa \in \{1,2\}$. Let $p$ be the floating point arithmetic precision (32 by default). Under the setting of Section 5.1 and Assumption 5.1, for any weight initialization satisfying $\|W_l^\epsilon(0)\| \leq \beta \; \forall l \in [L]$, w.p. $\geq 1 - \delta$ over sampling the dataset $(X,Y)$,*

$$R^\epsilon(t) \leq \hat{R}_\gamma^\epsilon(t) + \Upsilon_\kappa + \frac{\Delta_{\kappa,\beta}(t)\epsilon^\kappa}{\gamma}, \tag{6}$$

*for the terms in the rhs defined as*

$$\Upsilon_\kappa = \sqrt{\frac{\kappa p d \ln 2 + \ln(1/\delta)}{2m}}, \qquad \Delta_{\kappa,\beta}(t) = \Phi_\kappa v_\beta(t) u_\beta^{L-1}(t), \tag{7}$$

*where*

$$\Phi_\kappa = \begin{cases} L\sqrt{d} + 1 + (L-1)\rho, & \kappa = 1; \\ (L-1)\left[(L+1+(L-1)\rho)\sqrt{d} + 2(1+(L-1)\rho)\right], & \kappa = 2, \end{cases} \tag{8}$$

*where $\rho = \frac{\|W_1^\epsilon(0)\|_F}{\|W_1^\epsilon(0)\|}$ is the square root of the stable rank of the input layer at initialization, and the definitions of $u_\beta$ and $v_\beta$ are given below.*

*Define*

- $s = \left\|YX^\top \Sigma_X^{-1/2}\right\|; \quad \bar{1} = 1 + \rho\beta^L;$

- *For a given $r \geq 0$, $\bar{s}_r = (1 - \epsilon)(s + \rho\beta^L) + \epsilon\sqrt{r}(L-1)(1 + \rho\beta^L);$*

- $\bar{s} = \bar{s}_1; \quad \hat{s} = \frac{L}{1+(L-1)\rho}\bar{s}; \quad \bar{\beta} = \beta\frac{\bar{s}_\rho}{\bar{s}_1}.$

*We have*

$$u_\beta(t) = \begin{cases} \bar{\beta}e^{\bar{s}t}, & L = 2; \\ \left(\bar{\beta}^{2-L} - (L-2)\bar{s}t\right)^{\frac{1}{2-L}}, & L \geq 3; \end{cases} \tag{9}$$

$$v_\beta(t) = \frac{L-1}{L}\hat{s}^{\frac{2-L}{L}} u_\beta^{L-1}(t)[w(u_\beta(t)) - w(\beta)]e^{\frac{u_\beta^L(t)}{\hat{s}}}, \tag{10}$$

*where [1]*

$$w(u) = -\frac{\bar{1}}{\bar{s}}\Gamma\left(\frac{2-L}{L}, \frac{u^L}{\hat{s}}\right) - \rho\frac{\hat{s}}{\bar{s}}\Gamma\left(\frac{2}{L}, \frac{u^L}{\hat{s}}\right). \tag{11}$$

This theorem gives an *a priori* bound for the generalization gap $R^\epsilon - \hat{R}_\gamma^\epsilon$, i.e. it could be computed without performing the actual training.

**Dimension dependency.** Our bound does not depend on width $n_l \; \forall l \in [L-1]$, in contrast to the bounds of Dziugaite & Roy (2017); Zhou et al. (2019); Biggs & Guedj (2022). However, both penalty terms of Theorem 5.2, $\Upsilon_\kappa$ and $\Delta_{\beta,\kappa}(t)$, grow as $\sqrt{d}$ with the input dimension.

---

[1] $\Gamma$ is an upper-incomplete gamma-function defined as $\Gamma(s,x) = \int_x^\infty t^{s-1}e^{-t}\,dt$.

### 5.3 Choosing the training time

As we see, $\Delta_{\kappa,\beta}(t)$ diverges super-exponentially as $t \to \infty$ for $L = 2$ and as $t \to \frac{\bar{\beta}^{2-L}}{(L-2)\bar{s}}$ for $L \geq 3$, so the bound eventually becomes vacuous. For the bound to make sense, we should be able to find $t$ small enough for $\Delta_{\kappa,\beta}(t)$ to stay not too large, and at the same time, large enough for the training risk $\hat{R}_\gamma^\epsilon(t)$ to get considerably reduced.

In the present section, we are going to demonstrate that for values of $t$ which correspond to partially learning the dataset (i.e. for which $\hat{R}_\gamma^\epsilon(t) \in (0,1)$), the last term of the bound, $\Delta_{\kappa,\beta}(t)/\gamma$, admits a finite limit as $\beta \to 0$.[2]

We do not know how $\hat{R}_\gamma^\epsilon(t)$ decreases with $t$ in our case. However, when the network is linear, this can be computed explicitly when either the weight initialization is properly aligned with the data, or the initialization norm $\beta$ vanishes. We are going to perform our whole analysis for $L = 2$ in the main, and defer the case $L \geq 3$ to Appendix D.2.

That is, consider an SVD: $Y\tilde{X}^+ = \frac{1}{\sqrt{m}} Y X^\top (XX^\top)^{-1/2} = PSQ^\top$, where $P$ and $Q$ are orthogonal and $S$ is diagonal; note that $S_{11} = s$. Observe also that $s = \|Y\tilde{X}^+\| \leq \|Y\|\|\tilde{X}^+\| \leq 1$ since $y = \pm 1$. Saxe et al. (2013)[3] have demonstrated for linear nets ($\epsilon = 0$) and $L = 2$ that when the weight initialization is properly aligned with $P$ and $Q$,[4] $\|W_1^0(t)\| = \|W_2^0(t)\| = \bar{u}(t)$, where $\bar{u}$ satisfies[5]

$$\frac{d\bar{u}(t)}{dt} = \bar{u}(t)(s - \bar{u}^2(t)), \qquad \bar{u}(0) = \beta, \tag{12}$$

which gives the solution in implicit form:

$$t = \int \frac{d\bar{u}}{\bar{u}(s - \bar{u}^2)} = \frac{1}{2s} \ln\left(\frac{\bar{u}^2(t)(s - \beta^2)}{\beta^2(s - \bar{u}^2(t))}\right). \tag{13}$$

One could resolve $\bar{u}$ explicitly to get

$$\bar{u}(t) = \frac{se^{2st}}{e^{2st} - 1 + s/\beta^2}. \tag{14}$$

Saxe et al. (2013) observed this expression to hold with good precision for random (not aligned) initializations when $\beta$ is small enough. This is supported by further works (Li et al., 2021; Jacot et al., 2021): when a linear network is initialized close to the origin and the initial weights do not lie on a "bad" subspace, the gradient flow nearly follows the same trajectory as studied by Saxe et al. (2013).

Plugging $\bar{u}^2(t) = \alpha s$ into Equation (13), we get the time required for a linear network to learn a fraction $\alpha$ of the strongest mode:

$$t_\alpha^*(\beta) = \frac{1}{2s} \ln\left(\frac{\alpha(s - \beta^2)}{(1 - \alpha)\beta^2}\right). \tag{15}$$

Clearly, the learning time $t_\alpha^*(\beta)$ diverges whenever $\beta \to 0$, or $\alpha \to 1$.

Since $t_\alpha^*(\beta)$ is the time sufficient to learn a network for $\epsilon = 0$, we suppose it also suffices to learn a nonlinear network. Thus, we are going to evaluate our bound at $t = t_\alpha^*$. Since we need $\hat{R}_\gamma(t_\alpha^*) < 1$ for the bound to be non-vacuous, we should take $\gamma$ small relative to $\alpha$. We consider $\gamma = \alpha^\nu/q$ for $\nu, q \geq 1$.

Since the linear network learning time $t_\alpha^*(\beta)$ is correct for almost all initialization only when $\beta$ vanishes, we are going to work in the limit of $\beta \to 0$. Since we need $\alpha \in (\beta^2/s, 1)$, otherwise the linear training time is negative, we take $\alpha = \frac{r}{s}\beta^\lambda$ for $\lambda \in (0, 2]$ and $r > 1$.

---

[2]In the linear case ($\epsilon = 0$), this corresponds to the saddle-to-saddle regime of Jacot et al. (2021).

[3]Saxe et al. (2013) assumed $XX^\top = I_d$ and $\|YX^\top\| = s$, while not introducing the factor $\frac{1}{m}$ as we do in Equation (4). It is easy to see that the our gradient flow has exactly the same dynamics as the one studied by Saxe et al. (2013).

[4]That is, when $W_2(0) = P\bar{S}_2^{1/2}R^\top$, $W_1(0) = R\bar{S}_1^{1/2}Q^\top$, where $R$ is orthogonal, and $\bar{S}_1$ and $\bar{S}_2$ are constructed from $S$ by adding or removing zero rows and columns.

[5]Our $\bar{u}$ corresponds to $u^{1/(N_l - 1)}$ of Saxe et al. (2013).

Let us compute the quantities which appear in our bounds, at $t = t_\alpha^*(\beta)$:

$$u = \bar{\beta}\left(\frac{\alpha(s-\beta^2)}{(1-\alpha)\beta^2}\right)^{\frac{\bar{s}}{2s}} = \frac{\bar{s}_\rho}{\bar{s}_1} r^{\frac{\bar{s}}{2s}} \beta^{1+\frac{\bar{s}}{s}\left(\frac{\lambda}{2}-1\right)}(1+O(\beta)), \tag{16}$$

where we omitted the argument $t_\alpha^*(\beta)$ for brevity.

Since $\Gamma(0,x) = -\operatorname{Ei}(-x) = -\ln x + O_{x\to 0}(1)$ and $\Gamma(1,x) = e^{-x} = O_{x\to 0}(1)$, we have $w(\beta) = \frac{\bar{1}}{\bar{s}}\ln(\beta^2) + O(1)$. Similarly, whenever $1 + \frac{\bar{s}}{s}\left(\frac{\lambda}{2}-1\right) > 0$, we have $u = o(\beta)$, and $w(u) = \frac{\bar{1}}{\bar{s}}\ln\left(\beta^{2+\frac{\bar{s}}{s}(\lambda-2)}\right) + O(1)$.

Consider first $\lambda < 2$ and suppose $\nu = 1$. In this case, $w(u) - w(\beta) = \frac{\bar{1}}{s}\left(\frac{\lambda}{2}-1\right)\ln(\beta^2) + O(1)$, and

$$\frac{2uv}{\gamma} = \frac{\bar{s}_\rho^2}{\bar{s}_1^2}\frac{qr^{\frac{\bar{s}}{s}-1}\bar{1}}{\beta^{\left(\frac{\bar{s}}{s}-1\right)(2-\lambda)}}\left(\frac{\lambda}{2}-1\right)\ln(\beta^2)(1+O(1/\ln\beta)). \tag{17}$$

Since $\bar{s} \geq s$, this expression diverges as $\beta \to 0$. Note that we get the same order divergence even also for $1 + \frac{\bar{s}}{s}\left(\frac{\lambda}{2}-1\right) \leq 0$. Clearly, for $\nu > 1$, we get even faster divergence.

On the other hand, if $\lambda = 2$ then $w(u) - w(\beta) = \frac{\bar{1}}{\bar{s}}\ln\left(\frac{\bar{s}_\rho^2}{\bar{s}_1^2}r^{\frac{\bar{s}}{s}}\right) + O(\beta)$, and

$$\frac{2uv}{\gamma} = \frac{s^\nu\bar{s}_\rho^2}{\bar{s}_1^3}qr^{\frac{\bar{s}}{s}-\nu}\bar{1}\ln\left(\frac{\bar{s}_\rho^2}{\bar{s}_1^2}r^{\frac{\bar{s}}{s}}\right)\beta^{2(1-\nu)}(1+O(\beta)). \tag{18}$$

We get a finite limit only for $\nu = 1$, i.e. when $\gamma \propto \alpha$. In this case, the last term of the bound Equation (6) becomes

$$\frac{\Delta_{\kappa,\beta}\epsilon^\kappa}{\gamma} = \frac{\bar{1}s\bar{s}_\rho^2}{2\bar{s}_1^3}qr^{\frac{\bar{s}}{s}-1}\Lambda_\kappa\ln\left(\frac{\bar{s}_\rho^2}{\bar{s}_1^2}r^{\frac{\bar{s}}{s}}\right)\epsilon^\kappa(1+O(\beta)). \tag{19}$$

Summing up, we expect the empirical risk $\hat{R}_\gamma(t_\alpha^*)$ to be smaller than $\frac{1}{2}$ for large enough $q$ (i.e. for small $\gamma$ compared to $\alpha$), and the last term to stay finite as $\beta \to 0$ and vanish as $\epsilon^\kappa$. Therefore as long as $\Upsilon_\kappa$ is not large enough (i.e. when $\kappa pd$ is small compared to $m$), the overall bound becomes non-vacuous for small enough $\epsilon$ at least at the (linear) training time $t_\alpha^*$. We are going to evaluate our bound empirically in the upcoming section.

## 6 Experiments

**Setup.** We consider an $L$-layer bias-free fully-connected network of width 64 and train it to classify 0-4 versus 5-9 digits of MNIST (i.e. $m = 60000$). In order to approximate the gradient flow dynamics, we run gradient descent with learning rate 0.001. By default, we take $L = 2$, the floating point precision to be $p = 32$, downsample the images to 7x7, and initialize the layers randomly in a standard Pytorch way (plus, we rescale the weights to match the required layer norm $\beta$). For some experiments, we consider deeper networks, half-precision $p = 16$, downsample not so aggresively, or enforce the input layer weight matrix to have rank 1.

**Observations:**

- When we take $\kappa = 2$ and $\beta \leq 0.1$, the bound becomes nonvacuous up to $\epsilon = 0.001$ (Figure 1) and even up to $\epsilon = 0.01$ if we enforce $\rho = 1$ (Figure 3);

- We can have a non-vacuous bound also for 14x14 MNIST if we take $\epsilon = 0.001$, half-precision ($p = 16$), and enforce $\rho = 1$ (Figure 5); the bound slightly exceeds the random guess risk for the full-sized, 28x28 MNIST;

- The bound for $\kappa = 2$ is much stronger than that for $\kappa = 1$ (Figure 2);

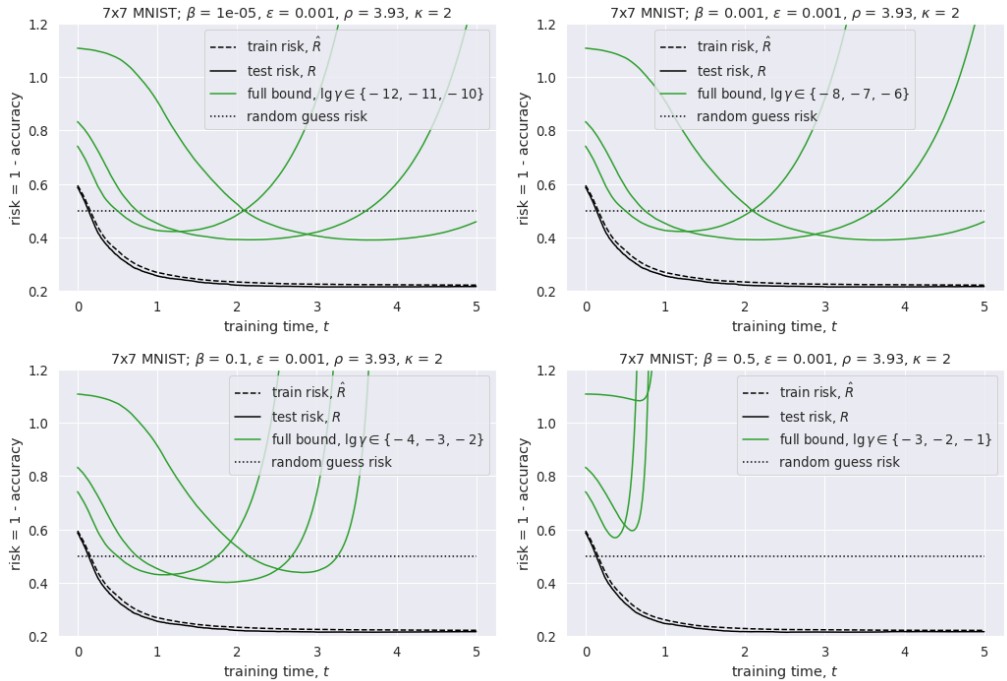

Figure 1: We consider 7x7 binary MNIST, $L = 2$, $\kappa = 2$, $\epsilon = 0.001$, and vary $\beta$. The bound of Theorem 5.2 converges as $\beta$ vanishes and increases as $\beta$ grows. The bound stays non-vacuous for a small enough $\beta$ and a properly choosen $\gamma$. We consider $\gamma = \beta^2/q$ for $q \in \{1, 10, 100\}$.

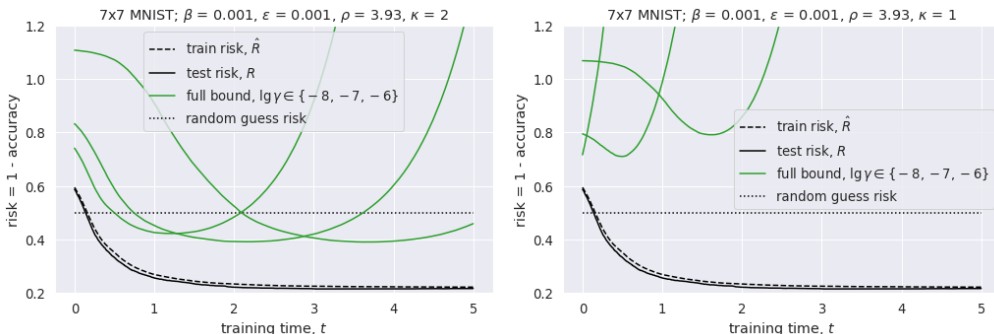

Figure 2: We consider 7x7 binary MNIST, $L = 2$, $\beta = 0.001$, $\epsilon = 0.001$, and compare different kappas of Theorem 5.2. The bound for $\kappa = 2$ is much stronger than that for $\kappa = 1$.

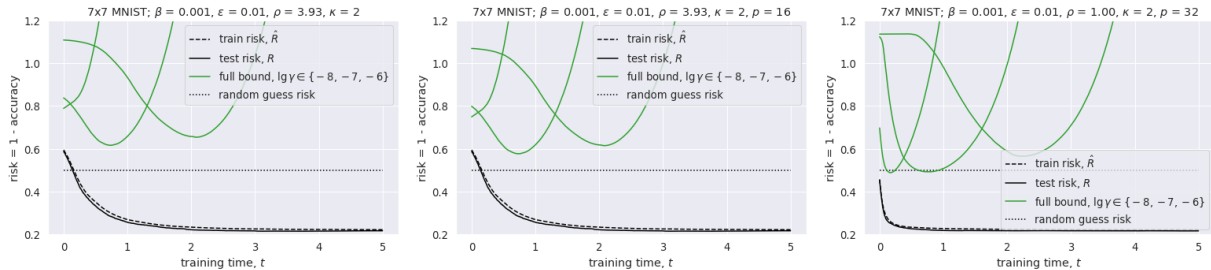

Figure 3: We consider 7x7 binary MNIST, $L = 2$, $\beta = 0.001$, $\epsilon = 0.01$, $\kappa = 2$, and vary the stable rank at initialization $\rho$ and floating point precision $p$. Initializing the input layer with a rank one matrix considerably improves the bound. Moreover, it also improves the convergence speed.

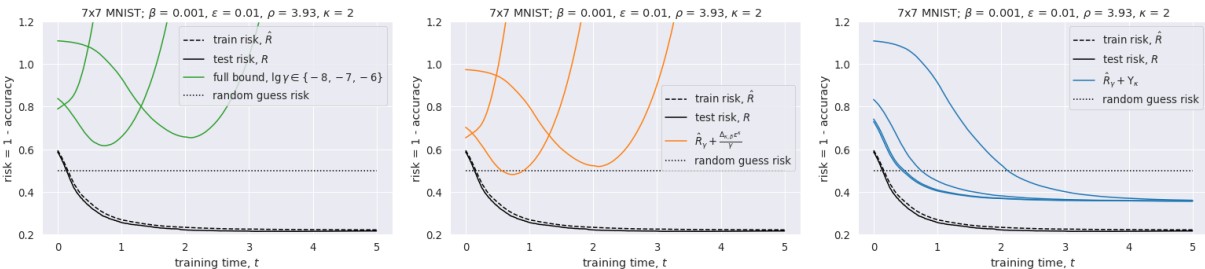

Figure 4: We consider 7x7 binary MNIST, $L = 2$, $\beta = 0.001$, $\epsilon = 0.01$, $\kappa = 2$, and compare different components of the bound. The left figure corresponds to the full bound, while for the central one we forget about the generalization gap bound for the proxy model $\Upsilon_\kappa$, and for the rightmost one, we forget about the deviation term $\frac{\Delta_{\kappa,\beta}\epsilon^\kappa}{\gamma}$. We see that both terms are of the same order; one therefore has to work on reducing both in order to reduce the overall bound.

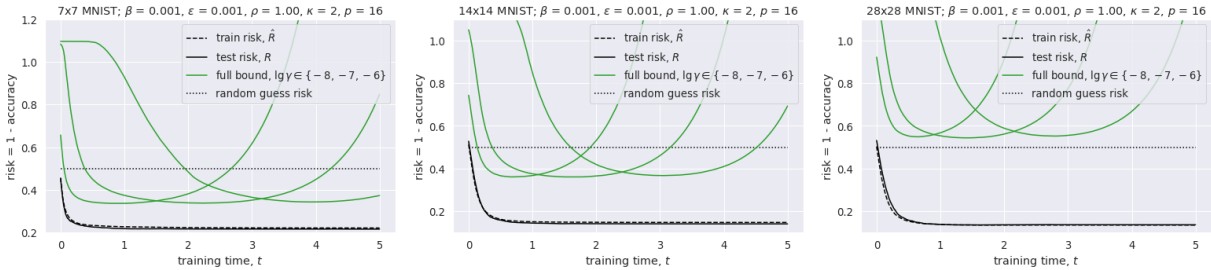

Figure 5: We consider binary MNIST, $L = 2$, $\beta = 0.001$, $\epsilon = 0.001$, $\kappa = 2$, $p = 16$, rank one input layer initialization, and vary image dimensions. In this "gentle" scenario, the bound stays non-vacuous for 14x14 MNIST, and only slightly exceeds the random guess risk for the full-sized, 28x28 MNIST.

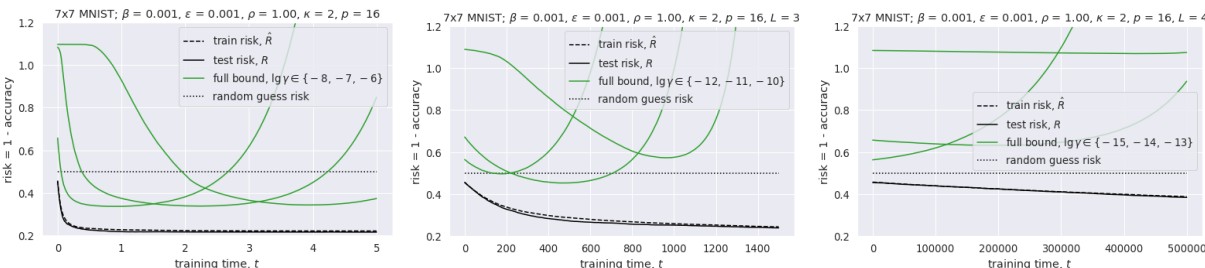

Figure 6: We consider binary 7x7 MNIST, $\beta = 0.001$, $\epsilon = 0.001$, $\kappa = 2$, $p = 16$, rank one input layer initialization, and vary depth. Even in this "gentle" scenario, the bound gets considerably worse with depth.

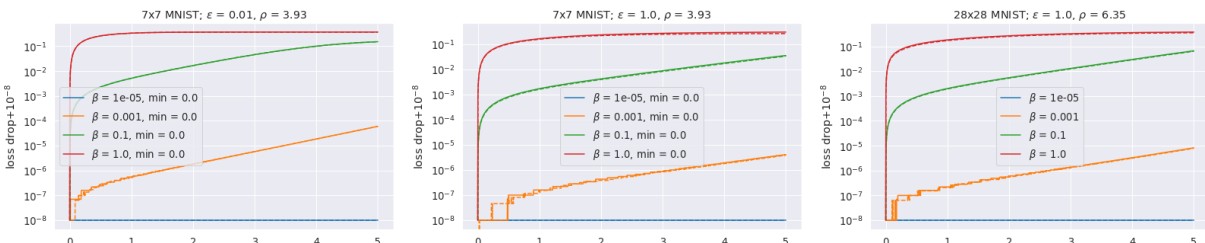

Figure 7: We consider binary MNIST, $L = 2$, $\beta = 0.001$, $\epsilon = 0.001$, and measure the differences $\mathcal{L}(0) - \mathcal{L}(t)$ (solid lines) and $\mathcal{L}_X(0) - \mathcal{L}_X(t)$ (dashed lines). Here $\mathcal{L}(t) = \frac{1}{2m} \left\| Y - f^\epsilon_{\theta^\epsilon(t)}(\tilde{X}) \right\|^2_F$ is the loss at time $t$, while $\mathcal{L}_X(t) = \frac{1}{2m} \left\| \left( Y - f^\epsilon_{\theta^\epsilon(t)}(\tilde{X}) \right) \tilde{X}^\top \right\|^2_F$ is the "projected" loss at time $t$. While $\mathcal{L}(t)$ should clearly decrease for gradient descent with small enough steps, it is not *a priori* clear that $\mathcal{L}_X(t)$ also does. As we see from the plots, it does for $\beta$ large and small, and for $\epsilon$ up to 1, which corresponds to conventional ReLU activations. These results validate our Assumption 5.1. Note that we added a small quantity $10^{-8}$ in order to make zero visible.

- The bound improves and converges as $\beta$ vanishes (Figure 1);

- Assumption 5.1 holds empirically (Figure 7);

- The bound improves if we enforce $\rho = 1$; this also results in faster convergence (Figure 3);

- The bound slightly improves for the half floating point precision (Figure 3);

- For $\epsilon = 0.01$ at the training time, all components of our bound are of the same order, while for larger $\epsilon$, the last one starts to dominate (Figure 4);

- The bound deteriorates considerably when we increase depth (Figure 6).

## 7 Proof of the main result

### 7.1 Proof of Theorem 5.2 for $\kappa = 1$

We start with approximating our learned network $f^\epsilon_{\theta^\epsilon}$ with $f^\epsilon_{\theta^0}$, i.e. with a nonlinear network that uses the weights learned by the linear one. This approximation deviates from the original network the following way: $\forall x \in \mathbb{R}^d, \epsilon \in [0, 1]$,

$$\frac{1}{\epsilon} \left\| f^\epsilon_{\theta^\epsilon}(x) - f^\epsilon_{\theta^0}(x) \right\| = \frac{1}{\epsilon} \left\| \int_0^\epsilon \frac{\partial f^\epsilon_{\theta^\tau}(x)}{\partial \tau} \, d\tau \right\| \leq \sup_{\tau \in [0,\epsilon]} \left\| \frac{\partial f^\epsilon_{\theta^\tau}(x)}{\partial \tau} \right\| \leq \sup_{\tau \in [0,\epsilon]} \sum_{l=1}^L \left\| \frac{\partial W^\tau_l}{\partial \tau} \right\| \prod_{k \neq l} \|W^\tau_k\| \|x\| \qquad (20)$$

since we use Leaky ReLUs. We omitted the argument $t$ for brevity. We get a similar deviation bound on the training dataset:

$$\frac{1}{\epsilon} \left\| f^\epsilon_{\theta^\epsilon}(\tilde{X}) - f^\epsilon_{\theta^0}(\tilde{X}) \right\| \leq \sup_{\tau \in [0,\epsilon]} \left\| \frac{\partial W^\tau_1 \tilde{X}}{\partial \tau} \right\|_F \prod_{k=2}^L \|W^\tau_k\| + \sup_{\tau \in [0,\epsilon]} \sum_{l=2}^L \left\| \frac{\partial W^\tau_l}{\partial \tau} \right\| \|W^\tau_1 \tilde{X}\|_F \prod_{k \in [2:L] \setminus \{l\}} \|W^\tau_k\|. \qquad (21)$$

We complete the above deviation bound with bounding weight norms and norms of weight derivatives:

**Lemma 7.1.** *Under Assumption 5.1, $\forall \tau \in [0, 1], t \geq 0$ $\frac{1}{\sqrt{m}} \|W^\tau_1(t)\tilde{X}\|_F \leq \rho u(t)$, $\frac{1}{\sqrt{m}} \left\| \frac{\partial W^\tau_1 \tilde{X}}{\partial \tau}(t) \right\|_F \leq v(t)$, and $\forall l \in [L]$ $\|W^\tau_l(t)\| \leq u(t)$, $\left\| \frac{\partial W^\tau_l}{\partial \tau}(t) \right\| \leq v(t)$ for $u$, $v$ defined in Theorem 5.2.*

See Section 7.3 and Appendix D.1 for the proof.

Now we can relate the risk of the original model with the risk of the approximation. Since $r \leq r_\gamma^C \leq r_\gamma$ and $r_\gamma^C(\cdot, y)$ is $1/\gamma$-Lipschitz for any fixed $y$,

$$
\begin{aligned}
R(f_{\theta^\epsilon}) - \hat{R}_\gamma(f_{\theta^\epsilon}) &\leq R_\gamma^C(f_{\theta^\epsilon}) - \hat{R}_\gamma^C(f_{\theta^\epsilon}) \\
&\leq R_\gamma^C(f_{\theta^0}) - \hat{R}_\gamma^C(f_{\theta^0}) + \frac{1}{\gamma}\mathbb{E}\left\| f_{\theta^\epsilon}^\epsilon(\tilde{x}) - f_{\theta^0}^\epsilon(\tilde{x}) \right\| + \frac{1}{\gamma m}\left\| f_{\theta^\epsilon}^\epsilon(\tilde{X}) - f_{\theta^0}^\epsilon(\tilde{X}) \right\|_1,
\end{aligned}
\tag{22}
$$

where the expectation is over the data distribution $\mathcal{D}$, and $\tilde{x} = \Sigma^{-1/2}x$ is the actual input of the network.

As for the last term, the deviation on the train dataset, we use that $\|z\|_1 \leq \sqrt{m}\|z\|$ for any $z \in \mathbb{R}^m$, and Equation (21). As for the deviation on the test dataset, due to Equation (20) and Lemma 7.1, in order to bound the last term, it suffices to bound $\mathbb{E}\|\tilde{x}\|$. Since $\Sigma = \mathbb{E}[xx^\top]$, we get

$$
\mathbb{E}\|\Sigma^{-1/2}x\|^2 = \mathbb{E}\left[x^\top\Sigma^{-1}x\right] = \mathbb{E}\,\mathrm{tr}\left[xx^\top\Sigma^{-1}\right] = \mathrm{tr}[I_d] = d.
\tag{23}
$$

This gives $\mathbb{E}\|\Sigma^{-1/2}x\| \leq \sqrt{\mathbb{E}\|\Sigma^{-1/2}x\|^2} \leq \sqrt{d}$.

The first two terms is a generalization gap of the proxy-model. We use a simple counting-based bound:

$$
R_\gamma^C\left(f_{\theta^0(t)}^\epsilon\right) - \hat{R}_\gamma^C\left(f_{\theta^0(t)}^\epsilon\right) \leq \sqrt{\frac{\ln|\mathcal{F}_1^\epsilon(t;\theta(0))| - \ln\delta}{2m}},
\tag{24}
$$

w.p. $\geq 1 - \delta$ over sampling the dataset $(X, Y)$, where $\mathcal{F}_1^\epsilon(t; \theta(0))$ denotes the set of functions representable with $f_{\theta^0(t)}^\epsilon$ for a given initial weights $\theta(0)$, where $\theta^0(t)$ is a result of running the gradient flow Equation (4) for time $t$. As long as we work with finite precision, this class is finite:

**Lemma 7.2.** $\forall\theta(0), \epsilon, t \geq 0, |\mathcal{F}_1^\epsilon(t; \theta(0))| \leq 2^{pd}$.

*Proof.* Since we run our gradient flow Equation (4) on whitened data to optimize squared loss, the initial weights are fixed, and the network we train is linear, the resulting weights depend only on $Y\tilde{X}^\top$ which has $d$ parameters. Since each function in our class is completely defined with the resulting weights, and each weight occupies $p$ bits, we get the above class size. $\square$

We could have used a classical VC-dimension-based bound instead (Vapnik & Chervonenkis, 1971). However, we found it to be numerically larger compared to the counting-based bound above.

This finalizes the proof of Theorem 5.2 for $\kappa = 1$.

## 7.2 Proof of Theorem 5.2 for $\kappa = 2$

What changes for $\kappa = 2$ is a proxy-model. Consider the following:

$$
\tilde{f}_{\theta^0, \theta^\epsilon}^\epsilon(x) = f_{\theta^0}^\epsilon(x) + \left(f_{\theta^\epsilon}^\epsilon(\tilde{X}) - f_{\theta^0}^\epsilon(\tilde{X})\right)\tilde{X}^+x.
\tag{25}
$$

That is, we take the same proxy-model as before, but we add a linear correction term. This correction term aims to fit the proxy model to the original one, $f_{\theta^\epsilon}^\epsilon$, on the training dataset $\tilde{X}$. We prove the following lemma in Appendix C.1:

**Lemma 7.3.** *Under the premise of Lemma 7.1, $\forall t \geq 0\ \forall\epsilon \in [0, 1]\ \forall x \in \mathbb{R}^d$ we have*

$$
\left\| f_{\theta^\epsilon(t)}^\epsilon(x) - \tilde{f}_{\theta^0(t), \theta^\epsilon(t)}^\epsilon(x) \right\| \leq (L-1)(L+1+\rho(L-1))u^{L-1}(t)v(t)\|x\|\epsilon^2;
\tag{26}
$$

$$
\left\| f_{\theta^\epsilon(t)}^\epsilon(\tilde{X}) - \tilde{f}_{\theta^0(t), \theta^\epsilon(t)}^\epsilon(\tilde{X}) \right\| \leq 2(L-1)(1+\rho(L-1))u^{L-1}(t)v(t)\sqrt{m}\epsilon^2.
\tag{27}
$$

The deviation now scales as $\epsilon^2$ instead of $\epsilon$.

What remains is to bound the size of $\mathcal{F}_2^\epsilon(t; \theta(0))$, which denotes the set of functions representable with our new proxy-model $\tilde{f}_{\theta^0(t), \theta^\epsilon(t)}^\epsilon$ for given initial weights $\theta(0)$. Since $\tilde{f}_{\theta^0(t), \theta^\epsilon(t)}^\epsilon$ is a sum of $f_{\theta^0(t)}^\epsilon$ and a linear model, its size is at most $2^{pd}$ times larger:

$$|\mathcal{F}_2^\epsilon(t; \theta(0))| \le |\mathcal{F}_1^\epsilon(t; \theta(0))| 2^{pd} \le 2^{2pd}. \tag{28}$$

This finalizes the proof of Theorem 5.2 for $\kappa = 2$.

### 7.3 Proof of Lemma 7.1

Below, we are going to present the proof only for $L = 2$, and defer the case of $L \ge 3$ to Appendix D.1.

#### 7.3.1 Weight norms

Let us expand the weight evolution Equation (4):

$$\frac{dW_1^\tau}{dt} = \left[ D^\tau \odot W_2^{\tau, \top} \Xi^\tau \right] \tilde{X}^\top, \qquad \frac{dW_2^\tau}{dt} = \Xi^\tau \left[ D^\tau \odot W_1^\tau \tilde{X} \right]^\top, \tag{29}$$

where we define

$$\Xi^\tau = \frac{1}{m} \left( Y - W_2^\tau \left[ D^\tau \odot W_1^\tau \tilde{X} \right] \right), \tag{30}$$

and $D^\tau = (\phi^\epsilon)' \left( W_1^\tau \tilde{X} \right)$ is a $n \times m$ matrix with entries equal to 1 or $1 - \tau$. We express it as $D^\tau = (1 - \tau) 1_{n \times m} + \tau \Delta$, where $\Delta$ is a $n \times m$ 0-1 matrix.

Let us bound the evolution of weight norms:

$$\frac{d \|W_1^\tau\|}{dt} \le \left\| \frac{dW_1^\tau}{dt} \right\| \le (1 - \tau) \|W_2^\tau\| \|\Xi^\tau \tilde{X}^\top\| + \tau \left\| \Delta \odot W_2^{\tau, \top} \Xi^\tau \right\| \|\tilde{X}\|. \tag{31}$$

Since multiplying by a 0-1 matrix elementwise does not increase Frobenius norm, we get

$$\left\| \Delta \odot W_2^{\tau, \top} \Xi^\tau \right\| \le \left\| \Delta \odot W_2^{\tau, \top} \Xi^\tau \right\|_F \le \|W_2^\tau\| \|\Xi^\tau\|_F. \tag{32}$$

Noting that $\Xi^\tau$ and $\Xi^\tau \tilde{X}^\top$ are row matrices, this results in

$$\frac{d \|W_1^\tau\|}{dt} \le \left( (1 - \tau) \|\Xi^\tau \tilde{X}^\top\| + \tau \|\Xi^\tau\| \|\tilde{X}\| \right) \|W_2^\tau\|. \tag{33}$$

By a similar reasoning,

$$\frac{d \|W_2^\tau\|}{dt} \le (1 - \tau) \|\Xi^\tau \tilde{X}^\top\| \|W_1^\tau\| + \tau \|\Xi^\tau\| \|W_1^\tau \tilde{X}\|_F. \tag{34}$$

Consider the following system of ODEs:

$$\frac{dg_1(t)}{dt} = \bar{s}^2 g_2(t), \quad \frac{dg_2(t)}{dt} = g_1(t), \qquad g_1(0) = \beta \bar{s}_\rho; \quad g_2(0) = \beta. \tag{35}$$

We then make use of the following lemma which we prove in Appendix C.2:

**Lemma 7.4.** $\|\Xi_\tau\| = \|\Xi_\tau\|_F \le \frac{1}{\sqrt{m}} (1 + \rho \beta^L)$. If we additionally take Assumption 5.1 then $\|\Xi_\tau \tilde{X}^\top\| = \|\Xi_\tau \tilde{X}^\top\|_F \le s + \rho \beta^L$.

This lemma implies $g_1(t) \ge (1 - \tau)(s + \rho \beta^2) \|W_1^\tau(t)\| + \tau (1 + \rho \beta^2) \|W_1^\tau(t) \tilde{X}\|_F$ and $g_2(t) \ge \|W_2^\tau(t)\|$.

The above system of ODEs could be solved analytically:

$$g_1(t) = \beta \sqrt{\bar{s}_\rho - \bar{s}_1} \cosh \left( \bar{s} t + \tanh^{-1} \left( \frac{\bar{s}_1}{\bar{s}_\rho} \right) \right), \qquad g_2(t) = \sqrt{\bar{\beta}^2 - \beta^2} \sinh \left( \bar{s} t + \tanh^{-1} \left( \frac{\bar{s}_1}{\bar{s}_\rho} \right) \right), \tag{36}$$

where $\bar{\beta} = \beta \frac{\bar{s}_\rho}{\bar{s}_1}$. This gives the bound for the input layer weight norms:

$$\|W_1^\tau(t)\| \leq \beta + \bar{s} \int_0^t g_2(t)\, dt = \beta - \bar{\beta} + \sqrt{\bar{\beta}^2 - \beta^2} \cosh\left(\bar{s}t + \tanh^{-1}\left(\frac{\bar{s}_1}{\bar{s}_\rho}\right)\right). \tag{37}$$

For further analysis, we will need simpler-looking bounds. Consider a looser bound: $g_2(t) \leq u(t)$ and $g_1(t) \leq \bar{s}u(t)$, where

$$\frac{du(t)}{dt} = \bar{s}u(t), \qquad u(0) = \beta \frac{\bar{s}_\rho}{\bar{s}_1}. \tag{38}$$

This ODE solves as $u(t) = \bar{\beta}e^{\bar{s}t}$.

So, we have $\|W_2^\tau(t)\| \leq u(t)$. As for the input layer weights, we get

$$\|W_1^\tau(t)\| \leq \beta + \bar{s} \int_0^t u(t)\, dt = \beta - \bar{\beta} + u(t) \tag{39}$$

Similarly, $\frac{1}{\sqrt{m}}\|W_1^\tau(t)\tilde{X}\|_F = \beta\rho - \bar{\beta} + u(t)$. As we do not want additive terms, we bound from above: $\|W_1^\tau(t)\| \leq u(t)$ and $\frac{1}{\sqrt{m}}\|W_1^\tau(t)\tilde{X}\| \leq \rho u(t)$.

### 7.3.2 Norms of weight derivatives

In Appendix C.3, we follow the same logic to demonstrate that $\left\|\frac{dW_1^\tau}{d\tau}\right\|$, $\frac{1}{\sqrt{m}}\left\|\frac{dW_1^\tau}{d\tau}\tilde{X}\right\|_F$, and $\left\|\frac{dW_2^\tau}{d\tau}\right\|$ are all bounded by the same $v$ which satisfies

$$\frac{dv}{dt} = v\left[(1+\rho)u^2 + \bar{s}\right] + u\left[\bar{1} + \rho u^2\right], \quad v(0) = 0. \tag{40}$$

It is a linear ODE in $v(t)$, and $u(t)$ is given: $u(t) = \bar{\beta}e^{\bar{s}t}$. We solve it in Appendix C.4 to get $v(t)$ from Theorem 5.2.

## 8 Conclusion

We have derived a novel generalization bound for LekyReLU networks. Our bound could be evaluated before the actual training and does not depend on network width. Our bound becomes non-vacuous for partially-trained nets with activation functions close to being linear.

### Broader Impact Statement

This is a theoretical work studying generalization in neural nets; its main result is a form of a generalization guarantee. Having a good generalization guarantee is necessary to make the models we use in practice more reliable. We do not foresee any negative societal or ethical impact our research could potentially cause.

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

# A    Discussion

## A.1    Assumptions

**Whitened data.**    Overall, the assumption on whitened data is not necessary for a result similar to Theorem 5.2 to hold. We assume the data to be whitened for two reasons. First, it legitimates the choice of training time $t_\alpha^*(\beta)$ since it is based on the analysis of Saxe et al. (2013), which assumed the data to be whitened. If we dropped it, we could still evaluate the bound of Theorem 5.2 at $t = t_\alpha^*(\beta)$, but it would be less clear whether the trainining risk $\hat{R}_\gamma$ becomes already small by this time.

Second, we had to bound $\|\tilde{X}\|$ throughout the proof of Theorem 5.2 and $\|\tilde{X}^+\|$ in the proof of Lemma 7.3. For whitened data, these are simply $\sqrt{m}$ and $1/\sqrt{m}$, which is a clear dependence on $m$, making the final bound look cleaner. Otherwise, they would be random variables whose dependence on $m$ would be less obvious.

Third, if we considered training on the original dataset $(Y, X)$ instead of the whitened one, $(Y, \tilde{X})$, we would have to know $YX^\top$ and $XX^\top$ in order to determine $\theta^0(t)$ for a given $t$ and initialization $\theta^0(0)$. These two matrices have $d + \frac{d(d+1)}{2}$ parameters, compared to just $d$ for $Y\tilde{X}^\top$. This way, $\Upsilon_\kappa$ would grow as $d$ instead of $\sqrt{d}$.

**Gradient flow.**    We expect our technique to follow through smoothly for finite-step gradient descent. Introducing momentum also seems doable. However, generalizing it to other training procedures, e.g. the ones which use normalized gradients, might pose problems since it is not clear how to bound the norm of the elementwise ratio of two matrices reasonably.

**Assumption 5.1.**    We use this assumption to prove the second part of Lemma 7.4. If we dropped it, the bound would be $\|\Xi^\tau \tilde{X}^\top\|_F \le \|\Xi^\tau\|_F \|\tilde{X}^\top\| \le 1 + \rho\beta^L$ instead of $s + \rho\beta^L$. This would result in larger exponents in the definition $u$ in Theorem 5.2.

As an argument in favor of this assumption, we demonstrate it empirically first (Figure 7), and we prove it for the linear case after, see Appendix E.

## A.2    Proof

We expect the bounds on weight norms $u(t)$ to be quite loose since we use Lemma 7.4 to bound the loss. This lemma bounds the loss with its value at the initialization, while the loss should necessarily decrease. If we could account for the loss decrease, the resulting $u(t)$ would increase with a lower exponent, or even stay bounded, as $\bar{u}(t)$, which corresponds to a linear model, does. This way, we would not have to assume $\epsilon$ to vanish as $\beta$ vanishes in order to keep the bound non-diverging for small $\beta$ at the training time $t_\alpha^*(\beta)$. Also, the general bound of Theorem 5.2 would diverge with training time $t$ much slower. We leave it for future work.

As we see from our estimates, $\Upsilon_\kappa$ becomes the main bottleneck of our bound for small $\epsilon$. The bound we used for $\Upsilon_\kappa$ is very naive; we believe that better bounds are possible.

Indeed, since the proxy corresponding to $\kappa = 1$ depends only on $\theta^0$, the trained linear model weights which one could obtain explicitly, this proxy, $f_{\theta^0}^\epsilon$, is also given explicitly. One could try to estimate its generalization gap explicitly, as done for kernel methods (Bordelon et al., 2024a) and for more complex models, which are closer to realisic neural nets (Bordelon et al., 2024b). We leave this direction for future work.

### A.3 Other architectures

As becomes apparent from the proof in Appendix C.1, the proxy-model for $\kappa = 2$, Equation (25), deviates from the original model $f_{\theta^\epsilon}^\epsilon$ as $O(\epsilon^2)$ for any map $(\epsilon, \theta, x) \to f_\theta^\epsilon(x)$ as long as the following holds:

1. $f_\theta^0(x)$ is linear in $x$ for any $\theta$;

2. $\frac{\partial^2 f_\theta^\epsilon(x)}{\partial \epsilon \partial \theta}$ is continuous as a function of $(\epsilon, \theta, x)$;

3. the result of learning $\theta^\epsilon(t)$ is differentiable in $\epsilon$ for any $t$.

This is directly applicable to convolutional networks with no other nonlinearities except for ReLU's; in partiular, without max-pooling layers. One may introduce max-poolings by interpolating between average-poolings (which are linear) for $\epsilon = 0$ and max-poolings for $\epsilon = 1$. This is not applicable to Transformers (Vaswani et al., 2017) since attention layers are inherently nonlinear: queries and keys have to be multiplied.

Compared to the fully-connected case of the present work, our bound might become even tighter for convolutional nets since $d$ becomes the number of color channels (up to 3) instead of the whole image size in pixels. However, the corresponding proxy-models might be over-simplistic: the linear net they will deviate from is just a global average-pooling followed by a linear $\mathbb{R}^d \to \mathbb{R}$ map. We leave exploring the convolutional net case for future work.

## B  How far could we get with our approach?

Recall our general proxy-based generalization bound of Equation (1):

$$R[f] \le \hat{R}_\gamma[f] + \left( R_\gamma^C[g] - \hat{R}_\gamma^C[g] \right) + \frac{1}{\gamma}\mathbb{E}_{x \sim \mathcal{D}}|f(x) - g(x)| + \frac{1}{\gamma}\frac{1}{m}\sum_{k=1}^m |f(x_k) - g(x_k)|. \tag{41}$$

Informally, we say that performance of $f$ is worse than that of $g$ at most by some deviation term.

The bound ends up to be good whenever (a) the generalization gap of $g$ could be well-bounded, and (b) $g$ does not deviate from $f$ much. That is why we considered proxy-models based on linear learned weights: their generalization gap could be easily bounded analytically and they do not deviate much from corresponding leaky ReLU nets as long as ReLU negative slopes are close to one.

The biggest conceptual disadvantage of this approach is that, given both $f$ and $g$ learn the training dataset, we have no chance proving that $f$ performs better than $g$, we could only prove that $f$ performs *not much worse* than $g$. Do the proxy-models we use in the present paper perform well, and how much do they deviate from original models? Our main theoretical result, Theorem 5.2, bounds the proxy-model generalization gap and the deviation from above. These bounds are arguably not optimal. It is therefore instructive to examine how well the bound would perform if we could estimate Equation (41) exactly.

### B.1 Empirical validation

**Setup.**   We work under the same setup as in Section 6[6], but instead of evaluating the bound of Theorem 5.2, we actually train a linear model with exactly the same procedure as for the original model, in order to get

---

[6]We also downsample MNIST images to 14x14 instead of 7x7. The reason why we do it is that on one hand, we wanted to test our bounds on more realistic scenarios, while on the other, $X$ does not appear to be full-rank for the original 28x28 MNIST.

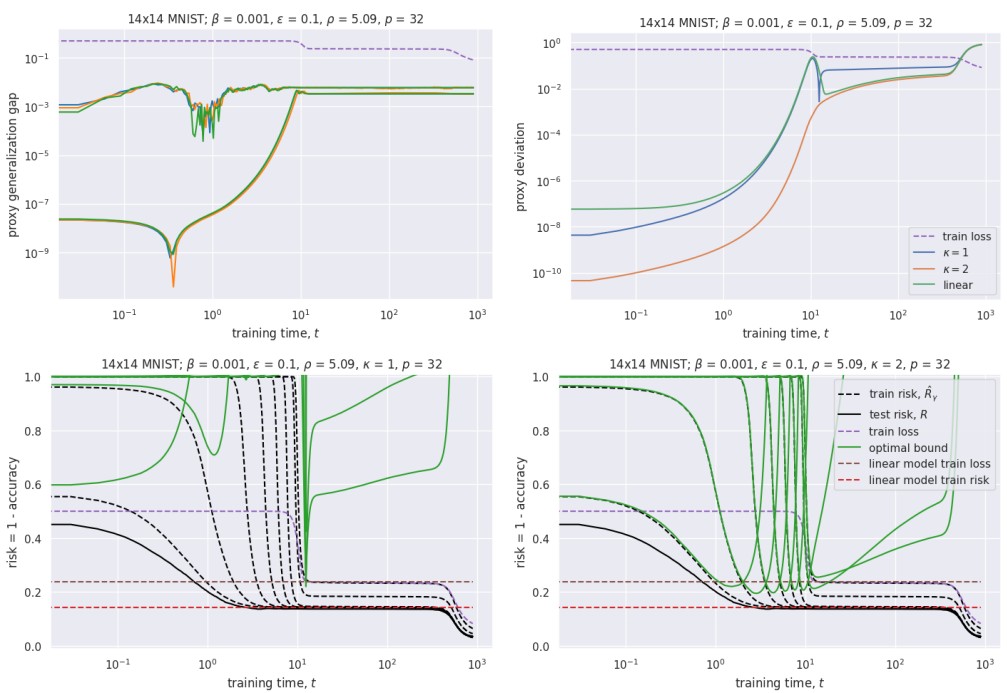

Figure 8: We examine the optimistic bound of Equation (41) for the proxy-models proposed in our paper: the $\kappa = 1$ one, $f_{\theta^0}^\epsilon$, the $\kappa = 2$ one of Equation (25), and also a linear proxy, $f_{\theta^0}^0$. The bottom row demonstrates the full bound (green lines), while the top one depicts the two components of the bound, namely, the proxy model generalization gap $R_\gamma^C(g) - \hat{R}_\gamma^C(g)$ and the proxy model deviation $\mathbb{E}\,|f(x) - g(x)| + \hat{\mathbb{E}}\,|f(x) - g(x)|$, separately. Different lines of the same color (e.g. solid green and dashed black lines on the bottom row) correspond to different values of $\gamma$. Proxy generalization gap stays low during the whole training (top left figure), while the train risk and the model deviation over gamma contribute significantly (bottom row, two groups of lines correspond to the minimal and the maximal $\gamma$ we considered). The optimistic bound for the 1st-order proxy (bottom left) gets non-vacuous only at the moment when GF escapes the origin and reaches the linear model loss. The bound for the 2nd-order proxy (bottom right) becomes non-vacuous soon after the original model becomes non-vacuous (but still stays near the origin), and stays non-vacuous until the model starts exploiting its nonlinearity to reduce the loss below the optimal linear model loss level (the last drop of purple and black lines).

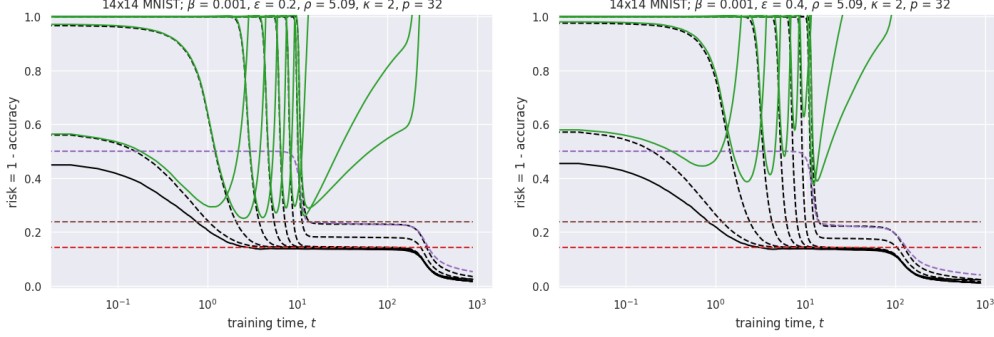

Figure 9: The optimistic bound of Equation (41) based on our $\kappa = 2$ proxy stays non-vacuous up to $\epsilon = 0.4$ until the gradient flow starts "exploiting" the nonlinearity (the last drop of purple and black lines).

trained linear weights $\theta^0$. We then evaluate the proxy-models considered in the present work: (1) the one for $\kappa = 1$, $f_{\theta^0}^\epsilon$, (2) the one for $\kappa = 2$, see Equation (25), and also (3) the linear network, $f_{\theta^0}^0$. We then evaluate the rhs of Equation (41) using a test part of the MNIST dataset. For this "optimistic" bound, we consider much larger values of epsilon: $\epsilon \geq 0.1$, i.e. our model much less "nearly-linear" now than before.

**Figures.** We present the results on Figures 8 and 9. Dashed lines correspond to the train set, while solid ones correspond to the test set. Black lines are risks of the actual trained model $f_{\theta^\epsilon(t)}^\epsilon$: $R(t)$ and $\hat{R}_\gamma(t)$, respectively. Green lines are our "optimistic" bound Equation (41) evaluated at different values of $\gamma$. Purple lines denote MSE loss of the actual trained model.

We also put three baselines on the plots. The dotted black line is the classification risk (and the MSE loss) of a zero model $f \equiv 0$. The brown dashed line is the MSE (train) loss of the optimal linear model, $f : x \to Y\tilde{X}^+ x$. Finally, the red dashed line is the (train) classification risk of the optimal linear model.

**Training phases.** As we observe on risk plots (Figure 9 and the bottom row of Figure 8), the training process could be divided into three phases. During the first phase, the risk decreases until it reaches the risk of the optimal linear model, while the loss stays at the level of $f \equiv 0$. This indicates that while the weights stay very close to the origin, the network outputs already align with the outputs of the optimal linear model. During the second phase, both the loss and the risk stay at the level of the optimal linear model. As for the following phase, both the risk and the loss drop below the optimal linear model level. Therefore from this point on, the network starts to "exploit" its nonlinearity in order to reduce the train loss.

**Observations:**

- The generalization gap stays negligible for all models and $\gamma$'s considered (Figure 8, top left);

- The proxy-model for $\kappa = 2$ approximates the original model best among all three proxies considered (Figure 8, top right);

- While the linear approximation deviates from the original model more than the one for $\kappa = 2$ during the first phase, their deviations are similar during the subsequent phases;

- At the same time, the $\kappa = 1$ approximation deviates more than that of $\kappa = 2$ during the second phase;

- The transition between the first and the second phases results in a nonmonotonic behavior of the deviation from the original model for $\kappa = 1$ and linear proxy-models;

- The resulting optimistic bound for $\kappa = 2$ (green lines of Figure 8, bottom right) stays non-vacuous during the first two phases for $\epsilon = 0.1$, while this is not the case for $\kappa = 1$ (green lines of Figure 8, bottom left);

- The optimistic bound for $\kappa = 2$ stays non-vacuous up to $\epsilon = 0.4$ (green lines of Figure 9).

It is tempting to assume that the weights $\theta^\epsilon$ follow the same trajectory as the weights of the linear model, $\theta^0$, during the first two phases. However, if it was the case, the $\kappa = 1$ proxy-model, $f_{\theta^0}^\epsilon$, would coincide with the original one, $f_{\theta^\epsilon}^\epsilon$, during this period. Then their quality would be the same; however, Figure 8, top right, demonstrates the opposite.

## C  Missing calculations in Section 7

### C.1  Proof of Lemma 7.3

We have:

$$\frac{1}{\epsilon^2}\left\|f_{\theta^\epsilon}^\epsilon(x) - \tilde{f}_{\theta^0,\theta^\epsilon}^\epsilon(x)\right\| = \frac{1}{\epsilon^2}\left\|\int_0^\epsilon \left(\frac{\partial f_{\theta^\tau}^\epsilon(x)}{\partial \tau} - \frac{\partial f_{\theta^\tau}^\epsilon(\tilde{X})\tilde{X}^+ x}{\partial \tau}\right) d\tau\right\| \leq \frac{1}{\epsilon}\sup_{\tau \in [0,\epsilon]}\left\|\frac{\partial f_{\theta^\tau}^\epsilon(x)}{\partial \tau} - \frac{\partial f_{\theta^\tau}^\epsilon(\tilde{X})}{\partial \tau}\tilde{X}^+ x\right\|.$$

$$(42)$$

Since $f_{\theta^\tau}^0$ is a linear network and rk $\tilde{X} = d$, we have $f_{\theta^\tau}^0(x) = f_{\theta^\tau}^0(\tilde{X})\tilde{X}^+ x$ and $\frac{\partial f_{\theta^\tau}^0(x)}{\partial \tau} = \frac{\partial f_{\theta^\tau}^0(\tilde{X})\tilde{X}^+ x}{\partial \tau}$. This implies

$$\frac{1}{\epsilon}\left\|\frac{\partial f_{\theta^\tau}^\epsilon(x)}{\partial \tau} - \frac{\partial f_{\theta^\tau}^\epsilon(\tilde{X})}{\partial \tau}\tilde{X}^+ x\right\| = \frac{1}{\epsilon}\left\|\int_0^\epsilon \left(\frac{\partial^2 f_{\theta^\tau}^\rho(x)}{\partial \rho \partial \tau} - \frac{\partial^2 f_{\theta^\tau}^\rho(\tilde{X})\tilde{X}^+ x}{\partial \rho \partial \tau}\right) d\rho\right\|$$

$$\leq \sup_{\rho \in [0,\epsilon]}\left\|\frac{\partial^2 f_{\theta^\tau}^\rho(x)}{\partial \rho \partial \tau} - \frac{\partial^2 f_{\theta^\tau}^\rho(\tilde{X})}{\partial \rho \partial \tau}\tilde{X}^+ x\right\| \tag{43}$$

$$\leq \sup_{\rho \in [0,\epsilon]}\left\|\frac{\partial^2 f_{\theta^\tau}^\rho(x)}{\partial \rho \partial \tau}\right\| + \sup_{\rho \in [0,\epsilon]}\left\|\frac{\partial^2 f_{\theta^\tau}^\rho(\tilde{X})}{\partial \rho \partial \tau}\right\|\|\tilde{X}^+ x\|.$$

Since we use LeakyReLU,

$$\left\|\frac{\partial^2 f_{\theta^\tau}^\rho(x)}{\partial \rho \partial \tau}\right\| \leq (L-1)\sum_{l=1}^L \left\|\frac{\partial W_l^\tau}{\partial \tau}\right\|\prod_{k \neq l}\|W_k^\tau\|\|x\|; \tag{44}$$

$$\left\|\frac{\partial^2 f_{\theta^\tau}^\rho(\tilde{X})}{\partial \rho \partial \tau}\right\| \leq (L-1)\left\|\frac{\partial W_1^\tau \tilde{X}}{\partial \tau}\right\|_F \prod_{k \in [2,L]}\|W_k^\tau\| + (L-1)\|W_1^\tau \tilde{X}\|_F \sum_{l=2}^L \left\|\frac{\partial W_l^\tau}{\partial \tau}\right\|\prod_{k \in [2,L]\setminus\{l\}}\|W_k^\tau\|. \tag{45}$$

Finally, since $\tilde{X}\tilde{X}^\top = mI_d$, we have $\|\tilde{X}^+\| = \frac{1}{\sqrt{m}}$. Combining everything together, we arrive into

$$\frac{\left\|f_{\theta^\epsilon}^\epsilon(x) - \tilde{f}_{\theta^0,\theta^\epsilon}^\epsilon(x)\right\|}{(L-1)\epsilon^2\|x\|} \leq \sup_{\tau \in [0,\epsilon]}\left[\sum_{l=1}^L \left\|\frac{\partial W_l^\tau}{\partial \tau}\right\|\prod_{k \neq l}\|W_k^\tau\|\right]$$

$$+ \sup_{\tau \in [0,\epsilon]}\left[\frac{1}{\sqrt{m}}\|W_1^\tau \tilde{X}\|_F \sum_{l=2}^L \left\|\frac{\partial W_l^\tau}{\partial \tau}\right\|\prod_{k \in [2,L]\setminus\{l\}}\|W_k^\tau\|\right] \tag{46}$$

$$+ \sup_{\tau \in [0,\epsilon]}\left[\frac{1}{\sqrt{m}}\left\|\frac{\partial W_1^\tau \tilde{X}}{\partial \tau}\right\|_F \prod_{k \in [2,L]}\|W_k^\tau\|\right].$$

Plugging the bounds from Lemma 7.1 then gives

$$\left\|f_{\theta^\epsilon}^\epsilon(x) - \tilde{f}_{\theta^0,\theta^\epsilon}^\epsilon(x)\right\| \leq (L-1)(L+1+(L-1)\rho)u^{L-1}v\|x\|\epsilon^2. \tag{47}$$

By a similar reasoning,

$$\frac{\left\|f_{\theta^\epsilon}^\epsilon(\tilde{X}) - \tilde{f}_{\theta^0,\theta^\epsilon}^\epsilon(\tilde{X})\right\|}{(L-1)\epsilon^2} \leq 2\sup_{\tau \in [0,\epsilon]}\left[\|W_1^\tau \tilde{X}\|_F \sum_{l=2}^L \left\|\frac{\partial W_l^\tau}{\partial \tau}\right\|\prod_{k \in [2,L]\setminus\{l\}}\|W_k^\tau\|\right]$$

$$+ 2\sup_{\tau \in [0,\epsilon]}\left[\left\|\frac{\partial W_1^\tau \tilde{X}}{\partial \tau}\right\|_F \prod_{k \in [2,L]}\|W_k^\tau\|\right]. \tag{48}$$

Plugging the same bounds,

$$\left\|f_{\theta^\epsilon}^\epsilon(\tilde{X}) - \tilde{f}_{\theta^0,\theta^\epsilon}^\epsilon(\tilde{X})\right\| \leq 2(L-1)(1+(L-1)\rho)u^{L-1}v\epsilon^2\sqrt{m}. \tag{49}$$

## C.2 Proof of Lemma 7.4

Recall the definition of $\Xi^\tau$:

$$\Xi^\tau(t) = \frac{1}{m}\left(Y - W_L^\tau(t)\left[D_{L-1}^\tau(t) \odot W_{L-1}^\tau(t)\left[\ldots W_2^\tau(t)\left[D_1^\tau(t) \odot W_1^\tau(t)\tilde{X}\right]\right]\right]\right). \tag{50}$$

Since $m\|\Xi^\tau\|_F^2$ is the loss function we optimize, it does not increase under gradient flow. Hence, since multiplying by $D_l^\tau$ elementwise does not increase Frobenius norm,

$$m\|\Xi^\tau(t)\|_F \le m\|\Xi^\tau(0)\|_F \le \|Y\|_F + \|W_1^\tau(0)\tilde{X}\|_F \prod_{l=2}^L \|W_l^\tau(0)\|. \tag{51}$$

We have $\|W_1^\tau(0)\tilde{X}\|_F = \rho\|W_1^\tau(0)\tilde{X}\| \le \|W_1^\tau(0)\|\|\tilde{X}\|$. Since all $y$ from $\operatorname{supp}\mathcal{D}$ are $\pm 1$ and $\tilde{X}\tilde{X}^\top = mI_d$, we get $\|\Xi^\tau(t)\|_F = \frac{1}{\sqrt{m}}(1 + \rho\beta^L)$.

Due to Assumption 5.1, $m\|\Xi^\tau \tilde{X}^\top\|_F^2$ does not increase under gradient flow:

$$m\|\Xi^\tau(t)\tilde{X}^\top\|_F \le m\|\Xi^\tau(0)\tilde{X}^\top\|_F \le \|Y\tilde{X}^\top\|_F + \|\tilde{X}\|\|W_1^\tau(0)\tilde{X}\|_F \prod_{l=2}^L \|W_l^\tau(0)\|. \tag{52}$$

Since $Y\tilde{X}^+ = s$, we have $Y\tilde{X}^\top = ms$. Therefore $\|\Xi^\tau(t)\tilde{X}^\top\|_F = s + \rho\beta^L$.

Finally, since $\Xi^\tau \in \mathbb{R}^{1\times m}$, $\|\Xi^\tau\| = \|\Xi^\tau\|_F$ and $\|\Xi^\tau \tilde{X}^\top\| = \|\Xi^\tau \tilde{X}^\top\|_F$.

### C.3 Bounding derivatives in the proof of Lemma 7.1

Let us start with $W_1^\tau$:

$$\frac{d^2 W_1^\tau}{dt\,d\tau} = \left[D^\tau \odot \left(\frac{dW_2^{\tau,\top}}{d\tau}\Xi^\tau + W_2^{\tau,\top}\frac{d\Xi^\tau}{d\tau}\right)\right]\tilde{X}^\top - \left[\bar{\Delta} \odot W_2^{\tau,\top}\Xi^\tau\right]\tilde{X}^\top; \tag{53}$$

$$\left\|\frac{d^2 W_1^\tau}{dt\,d\tau}\right\| \le \left\|\frac{dW_2^\tau}{d\tau}\right\|\left((1-\tau)\|\Xi^\tau \tilde{X}^\top\| + \tau\|\Xi^\tau\|_F\|\tilde{X}\|\right) + \|W_2^\tau\|\left(\left\|\frac{d\Xi^\tau}{d\tau}\right\|_F + \|\Xi^\tau\|_F\right)\|\tilde{X}\|. \tag{54}$$

We need a bound for $\left\|\frac{d\Xi^\tau}{d\tau}\right\|$:

$$m\frac{d\Xi^\tau}{d\tau} = W_2^\tau\left[\bar{\Delta} \odot W_1^\tau \tilde{X}\right] - W_2^\tau\left[D^\tau \odot \frac{dW_1^\tau}{d\tau}\tilde{X}\right] - \frac{dW_2^\tau}{d\tau}\left[D^\tau \odot W_1^\tau \tilde{X}\right]; \tag{55}$$

$$\begin{aligned}
m\left\|\frac{d\Xi^\tau}{d\tau}\right\| &\le \|W_2^\tau\|\|W_1^\tau \tilde{X}\|_F + \|W_2^\tau\|\left\|\frac{dW_1^\tau}{d\tau}\tilde{X}\right\|_F + \left\|\frac{dW_2^\tau}{d\tau}\right\|\|W_1^\tau \tilde{X}\|_F \\
&\le u^2\rho\sqrt{m} + u\left\|\frac{dW_1^\tau}{d\tau}\tilde{X}\right\|_F + u\rho\sqrt{m}\left\|\frac{dW_2^\tau}{d\tau}\right\|.
\end{aligned} \tag{56}$$

This results in

$$\frac{d}{dt}\left\|\frac{dW_1^\tau}{d\tau}\right\| \le \left\|\frac{d^2 W_1^\tau}{dt\,d\tau}\right\| \le u\left(1 + \rho\beta^2 + \rho u^2 + u\frac{1}{\sqrt{m}}\left\|\frac{dW_1^\tau}{d\tau}\tilde{X}\right\|_F\right) + \left\|\frac{dW_2^\tau}{d\tau}\right\|(s + (1-s)\tau + \rho\beta^2 + \rho u^2). \tag{57}$$

Similarly, we have an evolution of $W_1^\tau \tilde{X}$:

$$\begin{aligned}
\frac{d}{dt}\left\|\frac{dW_1^\tau}{d\tau}\tilde{X}\right\|_F &\le \left\|\frac{dW_2^\tau}{d\tau}\right\|\left((1-\tau)\|\Xi^\tau \tilde{X}^\top\|_F\|\tilde{X}\| + \tau\|\Xi^\tau\|_F\|\tilde{X}^\top \tilde{X}\|\right) + \|W_2^\tau\|\left(\left\|\frac{d\Xi^\tau}{d\tau}\right\|_F + \|\Xi^\tau\|_F\right)\|\tilde{X}^\top \tilde{X}\| \\
&\le u\left(1 + \rho\beta^2 + \rho u^2 + u\frac{1}{\sqrt{m}}\left\|\frac{dW_1^\tau}{d\tau}\tilde{X}\right\|_F\right)\sqrt{m} + \left\|\frac{dW_2^\tau}{d\tau}\right\|(s + (1-s)\tau + \rho\beta^2 + \rho u^2)\sqrt{m}.
\end{aligned} \tag{58}$$

Finally, consider $W_2^\tau$:

$$\frac{d^2 W_2^\tau}{dt\,d\tau} = \frac{d\Xi^\tau}{d\tau}\left[D^\tau \odot W_1^\tau \tilde{X}\right]^\top + \Xi^\tau\left[D^\tau \odot \frac{dW_1^\tau}{d\tau}\tilde{X} - \bar{\Delta} \odot W_1^\tau \tilde{X}\right]^\top; \tag{59}$$

$$\left\|\frac{d^2 W_2^\tau}{dt d\tau}\right\| \leq \left\|\frac{dW_1^\tau}{d\tau}\right\| \left((1-\tau)\|\Xi^\tau \tilde{X}^\top\| + \tau\|\Xi^\tau\|_F\|\tilde{X}\|\right) + \left(\|\Xi^\tau\| + \left\|\frac{d\Xi^\tau}{d\tau}\right\|\right)\|W_1^\tau \tilde{X}\|_F; \tag{60}$$

$$\frac{d}{dt}\left\|\frac{dW_2^\tau}{d\tau}\right\| \leq \left\|\frac{d^2 W_2^\tau}{dt d\tau}\right\|$$
$$\leq u\left(1 + \rho\beta^2 + \rho u^2 + u\frac{1}{\sqrt{m}}\left\|\frac{dW_1^\tau}{d\tau}\tilde{X}\right\|_F\right) + \left\|\frac{dW_1^\tau}{d\tau}\right\|(s + (1-s)\tau + \rho\beta^2) + \rho u^2\left\|\frac{dW_2^\tau}{d\tau}\right\|. \tag{61}$$

We see that $\left\|\frac{dW_1^\tau}{d\tau}\right\|$, $\frac{1}{\sqrt{m}}\left\|\frac{dW_1^\tau}{d\tau}\tilde{X}\right\|_F$, and $\left\|\frac{dW_2^\tau}{d\tau}\right\|$ are all bounded by the same $v$ which satisfies

$$\frac{dv(t)}{dt} = v(t)(\bar{s} + (1+\rho)u^2(t)) + u(t)(\bar{1} + \rho u^2(t)), \qquad v(0) = 0, \tag{62}$$

where $\bar{s} = s + (1-s)\tau + \rho\beta^2$ and $\bar{1} = 1 + \rho\beta^2$.

## C.4   Solving the ODE for $v(t)$

Recall $u(t) = \bar{\beta}e^{\bar{s}t}$. We solve the homogeneous equation to get

$$v(t) = C(t)e^{\bar{s}t + \frac{1+\rho}{2\bar{s}}\bar{\beta}^2 e^{2\bar{s}t}} = C(t)e^{(L-1)\ln u(t) + \frac{1+(L-1)\rho}{\bar{s}L}u^L(t)} = C(t)\bar{\beta}^{-1}u(t)e^{\frac{1+\rho}{2\bar{s}}u^2(t)}, \tag{63}$$

where $C(t)$ satisfies

$$\frac{dC(t)}{dt}u(t)e^{\frac{1+\rho}{2\bar{s}}u^2(t)} = \bar{\beta}u(t)\left[\bar{1} + \rho u^2(t)\right]. \tag{64}$$

Recall for $L = 2$, $\hat{s} = \frac{2}{1+\rho}\bar{s}$. Then

$$C(t) = \bar{\beta}\int e^{-\frac{u^2(t)}{\hat{s}}}\left[\bar{1} + \rho u^2(t)\right] dt$$
$$= \frac{\bar{\beta}}{2}\left[\frac{\bar{1}}{\bar{s}}\left(\text{Ei}\left(-\frac{u^2(t)}{\hat{s}}\right) - \text{Ei}\left(-\frac{\bar{\beta}^2}{\hat{s}}\right)\right) - \rho\frac{\hat{s}}{\bar{s}}\left(e^{-\frac{u^2(t)}{\hat{s}}} - e^{-\frac{\bar{\beta}^2}{\hat{s}}}\right)\right]. \tag{65}$$

This gives the final solution:

$$v(t) = \frac{1}{2}u(t)[w(u(t)) - w(\beta)]e^{\frac{u^2(t)}{\hat{s}}}, \tag{66}$$

where we took

$$w(u) = \frac{\bar{1}}{\bar{s}}\text{Ei}\left(-\frac{u^2(t)}{\hat{s}}\right) - \frac{2\rho}{1+\rho}e^{-\frac{u^2(t)}{\hat{s}}}. \tag{67}$$

# D   Deep networks

## D.1   Proof of Lemma 7.1 for $L \geq 3$

**Bounding weight norms.**   For $l \in [2, L]$,

$$\frac{dW_l^\tau}{dt} = \left[D_l^\tau \odot W_{l+1}^{\tau,\top} \ldots \left[D_{L-1}^\tau \odot W_L^{\tau,\top}\Xi^\tau\right]\right]\left[D_{l-1}^\tau \odot W_{l-1}^\tau \ldots \left[D_1^\tau \odot W_1^\tau\tilde{X}\right]\right]^\top; \tag{68}$$

$$\left\|\frac{dW_l^\tau}{dt}\right\| \leq \left((1-\tau)\|\Xi^\tau\tilde{X}^\top\|\|W_1^\tau\| + \tau(L-1)\|\Xi^\tau\|_F\|W_1^\tau\tilde{X}\|_F\right)\prod_{k\in[2,L]\setminus\{l\}}\|W_k^\tau\|. \tag{69}$$

For $l = 1$,

$$\frac{dW_1^\tau}{dt} = \left[D_1^\tau \odot W_2^{\tau,\top} \ldots \left[D_{L-1}^\tau \odot W_L^{\tau,\top}\Xi^\tau\right]\right]\tilde{X}^\top; \tag{70}$$

$$\left\| \frac{dW_1^\tau}{dt} \right\| \le \left( (1-\tau)\|\Xi^\tau \tilde{X}^\top\| + \tau(L-1)\|\Xi^\tau\|_F \|\tilde{X}\| \right) \prod_{k \in [2,L]} \|W_k^\tau\|; \tag{71}$$

$$\left\| \frac{dW_1^\tau \tilde{X}}{dt} \right\|_F \le \left( (1-\tau)\|\Xi^\tau \tilde{X}^\top\|_F \|\tilde{X}\| + \tau(L-1)\|\Xi^\tau\|_F \|\tilde{X}^\top \tilde{X}\| \right) \prod_{k \in [2,L]} \|W_k^\tau\|. \tag{72}$$

Using Lemma 7.4, we get $(1-\tau)\|\Xi^\tau \tilde{X}^\top\|\|W_1^\tau\| + \tau(L-1)\|\Xi^\tau\|\|W_1^\tau \tilde{X}\|_F \le g_1(t)$ and $\|W_l^\tau\| \le g_2(t) \; \forall l \in [2:L]$, where

$$\frac{dg_1(t)}{dt} = \bar{s}^2 g_2^{L-1}(t), \quad \frac{dg_2(t)}{dt} = g_1(t) g_2^{L-2}(t), \quad g_1(0) = \beta((1-\tau)(s+\beta^L) + \tau(L-1)(1+\beta^L)\rho), \quad g_2(0) = \beta. \tag{73}$$

We have the following first integral:

$$\frac{d}{dt}\left( g_1^2(t) - \bar{s}^2 g_2^2(t) \right) = 0, \quad g_1^2(0) - \bar{s}^2 g_2^2(0) = \beta^2 \left( ((1-\tau)(s+\beta^L) + \tau(L-1)(1+\beta^L)\rho)^2 - \bar{s}^2 \right). \tag{74}$$

Therefore

$$\frac{dg_2(t)}{dt} = g_2^{L-2}(t)\sqrt{\bar{s}^2 g_2^2(t) - \bar{s}^2 g_2^2(0) + g_1^2(0)}, \quad g_2(0) = \beta. \tag{75}$$

Suppose $\rho > 1$. Then $g_1^2(0) - \bar{s}^2 g_2^2(0) > 0$ and the solution is given in the following implicit form:

$$t = \frac{g_2^{3-L}(t)\, {}_2F_1\left( \frac{1}{2}, \frac{3-L}{2}, \frac{5-L}{2}, -\frac{\bar{s}^2 g_2^2(t)}{g_1^2(0) - \bar{s}^2 \beta^2} \right) - \beta_2^{3-L} F_1\left( \frac{1}{2}, \frac{3-L}{2}, \frac{5-L}{2}, -\frac{\bar{s}^2 \beta^2}{g_1^2(0) - \bar{s}^2 \beta^2} \right)}{(3-L)\sqrt{g_1^2(0) - \bar{s}^2 \beta^2}}. \tag{76}$$

The above expression cannot be made explicit for general $L$ (but we could get explicit expression for $L \in \{2, 3, 4\}$). As an alternative, we consider a looser bound $u(t)$:

$$\frac{du(t)}{dt} = \bar{s}_1 u^{L-1}(t), \quad u(0) = \beta \frac{\bar{s}_\rho}{\bar{s}_1}, \tag{77}$$

where $\bar{s}_\rho := (1-\tau)(s+\beta^L) + \tau(L-1)(1+\beta^L)\rho$; note that $\bar{s}_1 = \bar{s}$. We have $u(t) \ge g_2(t)$ and $\bar{s}u(t) \ge g_1(t)$ $\forall t \ge 0$. This ODE solves as

$$u(t) = \left( \bar{s}(2-L)t + \bar{\beta}^{2-L} \right)^{\frac{1}{2-L}}, \tag{78}$$

where $\bar{\beta} = \beta \frac{\bar{s}_\rho}{\bar{s}_1}$. Note that the solution exists only for $t < \frac{\bar{\beta}^{2-L}}{(L-2)\bar{s}}$.

For the input layer weights, we get

$$\frac{d\|W_1^\tau(t)\|}{dt} \le \bar{s}u^{L-1}(t). \tag{79}$$

The solution is given by

$$\|W_1^\tau(t)\| \le \beta + \bar{s}\int_0^t \left( \bar{s}(2-L)t + \bar{\beta}^{2-L} \right)^{\frac{L-1}{2-L}} dt = \beta - \bar{\beta} + \left( \bar{s}(2-L)t + \bar{\beta}^{2-L} \right)^{\frac{1}{2-L}} = \beta - \bar{\beta} + u(t). \tag{80}$$

Similarly, we have

$$\frac{1}{\sqrt{m}}\|W_1^\tau(t)\tilde{X}\|_F \le \beta\rho + \bar{s}\int_0^t \left( \bar{s}(2-L)t + \bar{\beta}^{2-L} \right)^{\frac{L-1}{2-L}} dt = \beta\rho - \bar{\beta} + u(t). \tag{81}$$

For brevity, we define

$$b = \beta - \bar{\beta} = \beta\left( 1 - \frac{\bar{s}_\rho}{\bar{s}_1} \right) = -\beta\tau(L-1)(\rho-1)\frac{1+\beta^L}{\bar{s}}, \tag{82}$$

and

$$b_\rho = \beta\rho - \bar{\beta} = \beta\left( \rho - \frac{\bar{s}_\rho}{\bar{s}_1} \right) = \beta(1-\tau)(\rho-1)\frac{s+\beta^L}{\bar{s}}. \tag{83}$$

As a simpler option, we could just say $\|W_1^\tau(t)\| \le u(t)$ and $\frac{1}{\sqrt{m}}\|W_1^\tau(t)\tilde{X}\| \le \rho u(t)$.

**Bounding norms of weight derivatives.** Recall the definition of $\Xi^\tau$:

$$\Xi^\tau = \frac{1}{m}\left(Y - W_L^\tau\left[D_{L-1}^\tau \odot W_{L-1}^\tau \dots \left[D_1^\tau \odot W_1^\tau \tilde{X}\right]\right]\right); \tag{84}$$

$$m\left\|\frac{d\Xi^\tau}{d\tau}\right\| \le (L-1)\|W_1^\tau\tilde{X}\|_F\prod_{l=2}^{L}\|W_l^\tau\| + \left\|\frac{dW_1^\tau\tilde{X}}{d\tau}\right\|\prod_{l=2}^{L}\|W_l^\tau\| + \|W_1^\tau\tilde{X}\|_F\sum_{k=2}^{L}\left\|\frac{dW_k^\tau}{d\tau}\right\|\prod_{l\in[2:L]\setminus\{k\}}\|W_l^\tau\|$$

$$\le u^{L-1}\left[(L-1)\sqrt{\rho m}u + \left\|\frac{dW_1^\tau\tilde{X}}{d\tau}\right\|_F + \sqrt{\rho m}\sum_{k=2}^{L}\left\|\frac{dW_k^\tau}{d\tau}\right\|\right]. \tag{85}$$

For $l \in [2, L]$,

$$\left\|\frac{d^2W_l^\tau}{dtd\tau}\right\| \le \left(\left\|\frac{d\Xi^\tau}{d\tau}\right\|_F + (L-1)\|\Xi^\tau\|_F\right)\|W_1^\tau\tilde{X}\|_F\prod_{k\in[2,L]\setminus\{l\}}\|W_k^\tau\|$$

$$+ \sum_{j\in[2:L]\setminus\{l\}}\left((1-\tau)\|\Xi^\tau\tilde{X}^\top\|\|W_1^\tau\| + \tau(L-1)\|\Xi^\tau\|_F\|W_1^\tau\tilde{X}\|_F\right)\left\|\frac{dW_j^\tau}{d\tau}\right\|\prod_{k\in[2,L]\setminus\{l,j\}}\|W_k^\tau\|$$

$$+ \left((1-\tau)\|\Xi^\tau\tilde{X}^\top\|\left\|\frac{dW_1^\tau}{d\tau}\right\| + \tau(L-1)\|\Xi^\tau\|_F\left\|\frac{dW_1^\tau\tilde{X}}{d\tau}\right\|_F\right)\prod_{k\in[2,L]\setminus\{l\}}\|W_k^\tau\| \tag{86}$$

$$\le \left(u^{L-1}\left[(L-1)\rho u + \frac{1}{\sqrt{m}}\left\|\frac{dW_1^\tau\tilde{X}}{d\tau}\right\|_F + \rho\sum_{k=2}^{L}\left\|\frac{dW_k^\tau}{d\tau}\right\|\right] + \bar{1}(L-1)\right)\rho u^{L-1}$$

$$+ \bar{s}u^{L-2}\sum_{j\in[2:L]\setminus\{l\}}\left\|\frac{dW_j^\tau}{d\tau}\right\| + \left((1-\tau)(s+\beta^L)\left\|\frac{dW_1^\tau}{d\tau}\right\| + \tau\bar{1}(L-1)\frac{1}{\sqrt{m}}\left\|\frac{dW_1^\tau\tilde{X}}{d\tau}\right\|_F\right)u^{L-2}.$$

For $l = 1$,

$$\left\|\frac{d^2W_1^\tau}{dtd\tau}\right\| \le \left(\left\|\frac{d\Xi^\tau}{d\tau}\right\|_F + (L-1)\|\Xi^\tau\|_F\right)\|\tilde{X}\|\prod_{k\in[2,L]}\|W_k^\tau\|$$

$$+ \sum_{j=2}^{L}\left((1-\tau)\|\Xi^\tau\tilde{X}^\top\| + \tau(L-1)\|\Xi^\tau\|_F\|\tilde{X}\|\right)\left\|\frac{dW_j^\tau}{d\tau}\right\|\prod_{k\in[2,L]\setminus\{j\}}\|W_k^\tau\|$$

$$\le \left(u^{L-1}\left[(L-1)\rho u + \frac{1}{\sqrt{m}}\left\|\frac{dW_1^\tau\tilde{X}}{d\tau}\right\|_F + \rho\sum_{k=2}^{L}\left\|\frac{dW_k^\tau}{d\tau}\right\|\right] + \bar{1}(L-1)\right)u^{L-1} + \bar{s}u^{L-2}\sum_{j=2}^{L}\left\|\frac{dW_j^\tau}{d\tau}\right\|. \tag{87}$$

$\left\|\frac{dW_l^\tau}{d\tau}\right\| \forall l \in [L]$, as well as $\frac{1}{\sqrt{m}}\left\|\frac{dW_1^\tau\tilde{X}}{d\tau}\right\|_F$, are all bounded by $v(t)$ which satisfies

$$\frac{dv}{dt} = \left[(L-1)\rho u^L + (1+(L-1)\rho)u^{L-1}v + (L-1)\bar{1}\right]u^{L-1} + \bar{s}(L-1)u^{L-2}v$$

$$= v\left[(1+(L-1)\rho)u^{2L-2} + \bar{s}(L-1)u^{L-2}\right] + (L-1)u^{L-1}\left[\bar{1} + \rho u^L\right], \qquad v(0) = 0, \tag{88}$$

where $\bar{1} = 1 + \beta^L$.

**Solving the ODE for v(t).** Recall

$$u(t) = \left(\bar{s}(2-L)t + \bar{\beta}^{2-L}\right)^{\frac{1}{2-L}}. \tag{89}$$

We solve the homogeneous equation to get

$$v(t) = C(t)e^{\frac{L-1}{2-L}\ln\left(\bar{s}(2-L)t+\bar{\beta}^{2-L}\right)+\frac{1+(L-1)\rho}{\bar{s}L}\left(\bar{s}(2-L)t+\bar{\beta}^{2-L}\right)^{\frac{L}{2-L}}}$$

$$= C(t)e^{(L-1)\ln u(t)+\frac{1+(L-1)\rho}{\bar{s}L}u^L(t)} = C(t)u^{L-1}(t)e^{\frac{1+(L-1)\rho}{\bar{s}L}u^L(t)}, \tag{90}$$

where $C(t)$ satisfies

$$\frac{dC(t)}{dt}u^{L-1}(t)e^{\frac{1+(L-1)\rho}{\bar{s}L}u^L(t)} = (L-1)u^{L-1}(t)\left[\bar{1}+\rho u^L(t)\right]. \tag{91}$$

Let us introduce $\hat{s} = \frac{L}{1+(L-1)\rho}\bar{s}$. Then

$$
\begin{aligned}
C(t) &= (L-1)\int e^{-\frac{u^L(t)}{\hat{s}}}\left[\bar{1}+\rho u^L(t)\right]\,dt \\
&= \frac{L-1}{L}\hat{s}^{\frac{2-L}{L}}\left[\frac{\bar{1}}{\bar{s}}\left(\Gamma\left(\frac{2-L}{L},\frac{\beta^L}{\hat{s}}\right)-\Gamma\left(\frac{2-L}{L},\frac{u^L(t)}{\hat{s}}\right)\right)+\rho\frac{\hat{s}}{\bar{s}}\left(\Gamma\left(\frac{2}{L},\frac{\beta^L}{\hat{s}}\right)-\Gamma\left(\frac{2}{L},\frac{u^L(t)}{\hat{s}}\right)\right)\right].
\end{aligned}
\tag{92}
$$

This gives the final solution:

$$v(t) = \frac{L-1}{L}\hat{s}^{\frac{2-L}{L}}u^{L-1}(t)[w(u(t))-w(\beta)]e^{\frac{u^L(t)}{\hat{s}}}, \tag{93}$$

where we took

$$w(u) = -\frac{\bar{1}}{\bar{s}}\Gamma\left(\frac{2-L}{L},\frac{u^L}{\hat{s}}\right) - \frac{L\rho}{1+(L-1)\rho}\Gamma\left(\frac{2}{L},\frac{u^L}{\hat{s}}\right). \tag{94}$$

## D.2 Evaluating the solution at the learning time

For a properly initialized linear network, $\forall l \in [L]$ $\|W_l^0(t)\| = \bar{u}(t)$, where $u(t)$ satisfies (Saxe et al., 2013)

$$\frac{d\bar{u}(t)}{dt} = \bar{u}^{L-1}(t)(s-\bar{u}^L(t)), \qquad \bar{u}(0) = \beta, \tag{95}$$

which gives the solution in implicit form[7]:

$$t_\alpha^*(\beta) = \int \frac{d\bar{u}}{\bar{u}^{L-1}(s-\bar{u}^L)} = \frac{\bar{u}^{2-L}(t)}{s(2-L)}\,{}_2F_1\left(1,\frac{2}{L}-1,\frac{2}{L};\frac{\bar{u}^L(t)}{s}\right) - \frac{\beta^{2-L}}{s(2-L)}\,{}_2F_1\left(1,\frac{2}{L}-1,\frac{2}{L};\frac{\beta^L}{s}\right). \tag{96}$$

Suppose we are going to learn a fixed fraction of the data, i.e. take $\bar{u}(t) = (\alpha s)^{1/L}$ for $\alpha \in (0,1)$. Then

$$t_\alpha^*(\beta) = \frac{\beta^{2-L}\,{}_2F_1\left(1,\frac{2}{L}-1,\frac{2}{L};\frac{\beta^L}{s}\right) - (\alpha s)^{\frac{2}{L}-1}\,{}_2F_1\left(1,\frac{2}{L}-1,\frac{2}{L};\alpha\right)}{s(L-2)}. \tag{97}$$

Since $t_\alpha^*(\beta)$ is the time sufficient to learn a network for $\epsilon = 0$, we suppose it also suffices to learn a nonlinear network. So, we are going to evaluate our bound at $t = t_\alpha^*$. Since we need $\hat{R}_\gamma(t_\alpha^*) < 1$ for the bound to be non-vacuous, we should take $\gamma$ small relative to $\alpha$. We consider $\gamma = \alpha^\nu/q$ for $\nu, q \geq 1$.

Since the linear network learning time $t_\alpha^*(\beta)$ is correct for almost all initialization only when $\beta$ vanishes, we are going to work in the limit of $\beta \to 0$. Since we need $\alpha \in (\beta^L/s, 1)$, otherwise the linear training time is negative, we take $\alpha = \frac{r}{s}\beta^\lambda$ for $\lambda \in (0, L)$ and $r > 1$.

Consider first $\lambda < L$:

$$t_\alpha^*(\beta) = \frac{\beta^{2-L}}{s(L-2)} + O(\beta^{2\lambda/L}). \tag{98}$$

Let us evaluate $u$ at this time:

$$u(t_\alpha^*(\beta)) = \left(\bar{\beta}^{2-L} - \frac{\bar{s}}{s}\beta^{2-L} + O(\beta^{2\lambda/L})\right)^{\frac{1}{2-L}} = \beta\left(\frac{\bar{s}_\rho^{2-L}}{\bar{s}_1^{2-L}} - \frac{\bar{s}}{s}\right)^{\frac{1}{2-L}}\left(1 + O\left(\beta^{L-2+2\lambda/L}\right)\right). \tag{99}$$

---

[7] ${}_2F_1$ is a hypergeometric function defined as a series ${}_2F_1(a,b,c,z) = 1+\sum_{k=1}^\infty \frac{(a)_k(b)_k}{(c)_k}\frac{z^k}{k!}$, where $(q)_k = q(q+1)\ldots(q+k-1)$.

Apparently, this expression does not make sense for $\rho = 1$ and even close to it, so we switch to $\lambda = L$, which we expect to be the right exponent:

$$t_\alpha^*(\beta) = \beta^{2-L} \frac{1 - r^{\frac{2}{L}-1}}{s(L-2)} + O(\beta^2).$$

(100)

Let us evaluate $u$ at this time:

$$u(t_\alpha^*(\beta)) = \left( \bar{\beta}^{2-L} - \frac{\bar{s}}{s}\left(1 - r^{\frac{2}{L}-1}\right)\beta^{2-L} + O(\beta^2) \right)^{\frac{1}{2-L}} = \beta\left( \frac{\bar{s}_\rho^{2-L}}{\bar{s}_1^{2-L}} - \frac{\bar{s}}{s}\left(1 - r^{\frac{2}{L}-1}\right) \right)^{\frac{1}{2-L}} \left(1 + O\left(\beta^L\right)\right).$$

(101)

This expression makes sense whenever $\frac{\bar{s}_\rho^{2-L}}{\bar{s}_1^{2-L}} - \frac{\bar{s}}{s}\left(1 - r^{\frac{2}{L}-1}\right) > 0$, i.e. when $r$ is close enough to 1.

Let us evaluate $w$ at the training time now. Since $\Gamma(a, x) = \Gamma(a) - \frac{x^a}{a} + O(x^{a+1})$, we get

$$w(u(t_\alpha^*(\beta))) - w(\beta) = \frac{\bar{1}}{\bar{s}} \frac{L}{2-L} \hat{s}^{\frac{L-2}{L}} \left( u^{2-L}(t_\alpha^*(\beta)) - \beta^{2-L} \right) + O(\beta^2).$$

$$= \frac{\bar{1}}{\bar{s}} \frac{L}{2-L} \hat{s}^{\frac{L-2}{L}} \beta^{2-L} \left( \frac{\bar{s}_\rho^{2-L}}{\bar{s}_1^{2-L}} - \frac{\bar{s}}{s}\left(1 - r^{\frac{2}{L}-1}\right) - 1 \right) + O(\beta^2).$$

(102)

Then the quantity of interest becomes

$$\frac{Lu^{L-1}(t_\alpha^*(\beta))v(t_\alpha^*(\beta))}{\gamma} = \frac{L-1}{\gamma} u^{2L-2}(t_\alpha^*(\beta)) \frac{\bar{1}}{\bar{s}} \frac{L}{2-L} \beta^{2-L} \left( \frac{\bar{s}_\rho^{2-L}}{\bar{s}_1^{2-L}} - \frac{\bar{s}}{s}\left(1 - r^{\frac{2}{L}-1}\right) - 1 \right)(1 + O(\beta^L))$$

$$= \frac{qs^\nu}{r^\nu} \frac{\bar{1}}{\bar{s}} \frac{L(L-1)}{2-L} \beta^{L(1-\nu)} \left( \frac{\bar{s}_\rho^{2-L}}{\bar{s}_1^{2-L}} - \frac{\bar{s}}{s}\left(1 - r^{\frac{2}{L}-1}\right) \right)^{\frac{2L-2}{2-L}} \left( \frac{\bar{s}_\rho^{2-L}}{\bar{s}_1^{2-L}} - \frac{\bar{s}}{s}\left(1 - r^{\frac{2}{L}-1}\right) - 1 \right)(1 + O(\beta^L)).$$

(103)

This expression does not diverge as $\beta \to 0$ when $\nu = 1$. We will also have a finite $\lim_{L\to\infty} \lim_{\beta\to 0}$ whenever $\epsilon \propto (L-1)^{-1}$.

## E  Proving Assumption 5.1 for linear nets

We have for $l = 1$,

$$\nabla_1 := \frac{1}{2m} \frac{\partial \left\| Y - f_{\theta^\epsilon}^\epsilon(\tilde{X}) \right\|_F^2}{\partial W_1^\epsilon} = -\left[ D_1^\epsilon \odot W_2^{\epsilon,\top} \dots \left[ D_{L-1}^\epsilon \odot W_L^{\epsilon,\top} \Xi^\epsilon \right] \right] \tilde{X}^\top.$$

(104)

For $l \in [2, L]$,

$$\nabla_l := \frac{1}{2m} \frac{\partial \left\| Y - f_{\theta^\epsilon}^\epsilon(\tilde{X}) \right\|_F^2}{\partial W_l^\epsilon} = -\left[ D_l^\epsilon \odot W_{l+1}^{\epsilon,\top} \dots \left[ D_{L-1}^\epsilon \odot W_L^{\epsilon,\top} \Xi^\epsilon \right] \right] \left[ D_{l-1}^\epsilon \odot W_{l-1}^\epsilon \dots \left[ D_1^\epsilon \odot W_1^\epsilon \tilde{X} \right] \right]^\top.$$

(105)

We also have

$$\nabla_1^X := \frac{1}{2m} \frac{\partial \left\| \left( Y - f_{\theta^\epsilon}^\epsilon(\tilde{X}) \right) \tilde{X}^\top \right\|_F^2}{\partial W_1^\epsilon} = -\left[ D_1^\epsilon \odot W_2^{\epsilon,\top} \dots \left[ D_{L-1}^\epsilon \odot W_L^{\epsilon,\top} \Xi^\epsilon \tilde{X}^\top \tilde{X} \right] \right] \tilde{X}^\top.$$

(106)

$$\nabla_l^X := \frac{1}{2m} \frac{\partial \left\| \left( Y - f_{\theta^\epsilon}^\epsilon(\tilde{X}) \right) \tilde{X}^\top \right\|_F^2}{\partial W_l^\epsilon}$$

$$= -\left[ D_l^\epsilon \odot W_{l+1}^{\epsilon,\top} \dots \left[ D_{L-1}^\epsilon \odot W_L^{\epsilon,\top} \Xi^\epsilon \tilde{X}^\top \tilde{X} \right] \right] \left[ D_{l-1}^\epsilon \odot W_{l-1}^\epsilon \dots \left[ D_1^\epsilon \odot W_1^\epsilon \tilde{X} \right] \right]^\top.$$

(107)

The statement of Assumption 5.1 follows from

**Conjecture E.1.** $\forall \epsilon \in [0,1] \; \forall t \geq 0 \; \sum_{l=1}^{L} \mathrm{tr}\left[\nabla_l^X \nabla_l^\top\right] \geq 0.$

Indeed, the above conjecture states that loss gradients wrt weights and "projected" loss gradients wrt weights are positively aligned, so, whenever loss does not increase, neither does projected loss. Since we use gradient flow, loss is guaranteed to not increase. Below, we prove the conjecture for linear nets.

*Proof of Conjecture E.1 for $\epsilon = 0$.* Since $\epsilon$ is zero, we omit the corresponding sup-index:

$$\mathrm{tr}\left[\nabla_1^X \nabla_1^\top\right] = \mathrm{tr}\left[\left[W_2^\top \ldots W_L^\top \Xi \tilde{X}^\top \tilde{X}\right] \tilde{X}^\top \tilde{X} \left[W_2^\top \ldots W_L^\top \Xi\right]^\top\right]$$

$$= \mathrm{tr}\left[\left[W_2^\top \ldots W_L^\top \Xi \tilde{X}^\top\right]\left[W_2^\top \ldots W_L^\top \Xi \tilde{X}^\top\right]^\top\right] = \mathrm{tr}\left[W_L \ldots W_2 W_2^\top \ldots W_L^\top\right] \mathrm{tr}\left[\Xi X^\top X \Xi^\top\right]$$

$$\tag{108}$$

by the circular property of trace, and the fact that $\Xi$ is a matrix with a single row. Since $d_{out} = 1$, both traces are just squared Euclidean norms of vectors, hence they are non-negative: $\mathrm{tr}\left[\nabla_1^X \nabla_1^\top\right] \geq 0.$

Let us do the same for the other layers:

$$\mathrm{tr}\left[\nabla_l^X \nabla_l^\top\right] = \mathrm{tr}\left[\left[W_{l+1}^\top \ldots W_L^\top \Xi \tilde{X}^\top \tilde{X}\right]\left[W_{l-1} \ldots W_1 \tilde{X}\right]^\top \left[W_{l-1} \ldots W_1 \tilde{X}\right]\left[W_{l+1}^\top \ldots W_L^\top \Xi\right]^\top\right]$$

$$= \mathrm{tr}\left[\left[W_{l+1}^\top \ldots W_L^\top \Xi \tilde{X}^\top W_1^\top \ldots W_{l-1}^\top\right]\left[W_{l+1}^\top \ldots W_L^\top \Xi \tilde{X}^\top W_1^\top \ldots W_{l-1}^\top\right]^\top\right] \tag{109}$$

$$= \mathrm{tr}\left[W_L \ldots W_{l+1} W_{l+1}^\top \ldots W_L^\top\right] \mathrm{tr}\left[\Xi X^\top W_1^\top \ldots W_{l-1}^\top W_{l-1} \ldots W_1 X \Xi^\top\right] \geq 0$$

for the same reasons as before. $\qquad\square$

Clearly, Conjecture E.1 should also hold for small enough $\epsilon$ whenever it holds for $\epsilon = 0$. However, the bound for the maximal $\epsilon$ for which we were able to guarantee the conjecture statement, vanishes with time $t$, as our weight bounds are too loose. For this reason, we do not include it here. See Section 6 for empirical validation of Assumption 5.1.

