# OpenReview forum: "A Generalization Bound for Nearly-Linear Networks"
_TMLR — Accepted by TMLR_

### Review · Reviewer_6Ktd · 2024-08-13

**Summary Of Contributions:**

The paper proposes a new form of generalization bound for deep networks that are *nearly* linear. The generalization bound roughly proceeds as follows: the authors demonstrate that nearly linear networks behave similarly to linear networks, then apply known generalization results for linear networks. The authors empirically demonstrate that the bound can be non-vacuous on real-world tasks for sufficiently linear networks.

**Audience:**

Yes

**Broader Impact Concerns:**

No broader impact concerns.

**Claims And Evidence:**

Yes

**Requested Changes:**

**Critical for acceptance**
- Please make the font sizes in the figures larger; they are not legible at this size

**Would strengthen**
- Do practical neural networks generalize well because of their similarity to linear networks or due to some other factor (I understand that a complete answer is outside of the scope of the paper)?
- Do other generalization bounds provide vacuous results on nearly linear networks?
- All practical neural networks close to nearly linear networks of similar size?
- Could the authors comment on the relationship between their work and NTK-type linear analysis (see above)?

**Strengths And Weaknesses:**

**Strengths**

At its core, the paper proposes a simple and powerful idea of leveraging the similarity of non-linear networks to linear networks to show their generalization ability. The theory is not only strong and well-motivated, it's also intuitively appealing. Experiments are also adequately conducted to back up the theory. Optimistically, bounds using the core idea of this paper could potentially produce the next generation of empirically-relevant generalization bounds.

Beyond technical aspects, the paper overall is generally well-written, with concepts presented at the appropriate depth.

Also, the authors are open about their bound's strengths and weaknesses, which is great to see.

**Weaknesses**

The primary weakness in my view is that the practical significance of the work remains unclear. It seems fairly obvious that if a nonlinear network is sufficiently close to a linear network, then a generalization bound for a linear network can be applied to the nonlinear network with some added error constant. The key to whether this kind of bound can be significant is the size of the error constant. Unfortunately, it's unclear whether the error constant derived by the authors should be considered large or small for practical networks. The authors show that with leaky ReLU with a slope of 0.99, the bound can be non-vacuous, but I'm not sure whether this provides much insight into most neural networks (which have significantly more non-linear activations).

Another way to phrase this is as an empirical question: do general nonlinear networks generalize well because of their similarity to linear networks or due to some other factor? The results in this paper hint at the exciting possibility that the answer might be yes, but the results merely suggest this rather than demonstrating a step in this direction. One way the authors could actually show this is by doing the following: take a nearly linear network and show that several other modern generalization bounds provide vacuous results on the network while the authors' proposed method provides a non-vacuous bound. Another approach is to show that practical neural networks can be approximated with nearly linear networks under a similar number of parameters (perhaps a constant multiple of the original network's parameter count).

Another question that I think will be important to address: wide neural networks in the neural tangent kernel (NTK) regime can be viewed as linear models in the limit of infinite width. Correspondingly, we may derive generalization bounds for wide neural networks as a function of their width (with the generalization bound approaching that of the linear model in the infinite-width limit). In this work, instead of infinite width, the authors consider the limit of infinitesimal nonlinearity to produce an alternate generalization bound based on another linear network. Could the authors comment on 1) the difference in the generalization bounds of the linear models in the two cases (I think the same parameter counting bound can't be applied in the NTK case), 2) the difference between the linear and nonlinear bound in the two cases as a function of width and activation nonlinearity level respectively?

Overall, while I'm not sure *how* significant this paper is yet, I'm inclined to believe that is more likely to be of interest to the community than not.

---

> ### Author Response · Authors · 2024-09-20
> **Response to Reviewer 6Ktd**
>
> We thank the anonymous reviewer for valuable comments!
>
> Concerning the questions posed:
>
> 1. **Do practical neural networks generalize well because of their similarity to linear networks or due to some other factor?**
>
> This must be false for data not explicable by a linear model during the late phase of training: in this case, the model has to exploit its nonlinearity to fit the data. On the other hand, this might be true in the beginning of training; note that the deviation term we derive monotonically grows with training time, but it is zero at $t=0$. In Section 4.3 we demonstrate that our bound stays non-vacuous up to $t \leq t_\alpha^*(\beta)$, the linear network training time, and small $\epsilon$.
>
> We emphasize, however, that the proxy models we use are not linear (as functions of $x$) themselves: they only use weights of a trained linear network. While the $\kappa=2$ proxy model approximates the original model provably better than just a linear model ($O(\epsilon^2)$ error vs $O(\epsilon)$), the approximation error still deviates with training time: we expect it to be far from the original model if we train up to convergence in our scenario.
>
> Nevertheless, being multilayer is an important component of feature learning: a model with only single layer of weights (e.g. a random feature model) cannot learn useful features from data. Indeed, the kernel associated to a parametric model, NTK, is constant if the model is linear in weights. A kernel being constant implies the features being constant. However, even multilayer linear networks (as our $f^0_\theta$) exhibit feature learning: see e.g. [1] for discussion.
>
> It is a very interesting question on its own what kind of features our proxy models learn, and does they help them to generalize.
>
> Also, in a low-data regime (out of the scope of the present work), even a linear model ($\epsilon = 0$) is able to fit the data. Hence the model does not necessarily need to "exploit" its nonlinearity. We therefore expect the model to stay close to being linear even up to convergence when $\epsilon$ is small enough; hence its generalization ability should be closely tied to the generalization of the linear network. We did not study this case in the present work because the counting-based bound $\Upsilon$ we use becomes vacuous when $d > m$. What we need is a generalization bound for our proxy models suitable for a high-dimensional, low-data regime. Since our proxy models have never appeared in the literature before, providing generalization bounds for them seems to be a new and exciting research direction. Alternatively, we could use a linear model $f^0_{\theta^0}$ as a proxy. For linear models, generalization bounds in both low- and high-dimensional cases exist in the literature: see e.g. [2], but they require strong assumptions on the data distribution (e.g. sub-Gaussian features and noisy linear teacher that generates $y$). The linear model demonstrates $O(\epsilon)$ deviation, similar to $f^\epsilon_{\theta^0}$, the proxy for $\kappa=1$, but the constant is larger. See Figure 8, top right, for comparison.
>
> 2. **Do other generalization bounds provide vacuous results on nearly linear networks?**
>
> As discussed in Section 2, non-vacuous bounds already exist, and we do not see any obstacles to applying them to nearly linear networks. However, as well also discuss there, none of the non-vacuous bounds we are aware of are a-priori. The only a-priori bounds we are aware of are uniform bounds, which are all vacuous for neural nets, and do not exploit near-linearity (and we see no way of making them $\epsilon$-aware). Overall, we are not aware of any existing bounds who decrease as $\epsilon$ becomes smaller; hence neither of them is able to exploit near-linearity in a fruitful way.
>
> 3. **All practical neural networks close to nearly linear networks of similar size?**
>
> This is the question of how much our proxy models deviate from original ones. As the deviation term we derive suggests, they are close at the beginning of training, and separate more and more as the training progresses. We investigate this question in Section B in the Appendix; see Figures 8 and 9 in particular.
>
> 4. **Could the authors comment on the relationship between their work and NTK-type linear analysis (see above)?**
>
> We discuss this question in our next comment below.
>
> [1] Tu et al., 2024: Mixed Dynamics In Linear Networks: Unifying the Lazy and Active Regimes https://arxiv.org/abs/2405.17580
>
> [2] Tsigler and Bartlett, 2023: Benign overfitting in ridge regression https://www.jmlr.org/papers/v24/22-1398.html

---

> > ### Author Response · Authors · 2024-09-20
> > **Answer to Question 4**
> >
> > See the previous comment above for the answers to the first three questions posed.
> >
> > Our answer coincides with the response given to the other reviewer asking a similar question.
> >
> > 4. **Could the authors comment on the relationship between their work and NTK-type linear analysis (see above)?**
> >
> > Concerning the suggested comparison with NTK, we believe there is a terminological confusion. Our proxy models use weights of a trained linear network, where "linear" means "without activation functions". That is, it is linear in its inputs, $x$, but not its weights. Whereas in the NTK limit, the model is trained as if it was linear in weights, $\theta$. Training of the model becomes equivalent to the training of a kernel method with NTK being a kernel. This kernel is not linear, hence the trained model is not linear in $x$.
> >
> > We emphasize that the proxy models we use, besides using weights of a trained linear model, are not linear themselves.
> >
> > We also emphasize that for the network to be close to its NTK limit, it should be either (a) very wide and parameterized in a certain (non-standard) way, or (b) initialized with very large weights (also not following the standard practice). Otherwise, there is still an associated kernel (also called NTK, but not to be confused with "limit NTK"), but the kernel is stochastic (since weights are initialized randomly) and evolves throughout training. Moreover, the evolution of NTK (not present in the NTK limit) is favorable as it is associated to feature learning. In our approach, we require neither special parameterization nor large width, while the associated kernel does evolve since the model is not linear in weights.
> >
> > Therefore our approach is quite orthogonal to NTK and generalizability of kernel methods. In terms of high-level methodology, the closest paper we could mention is [3], where they consider a two-layer NTK-parameterized MLP, and prove a generalization bound for it when width is sufficiently large. For this, they combine a generalization bound for the infinite-width limit when the kernel is constant, and a term that takes into account the fact that the kernel for finite-width deviates from the limit one. Whereas what we do is we combine a generalization bound for a proxy model that exploits the weights of a linear network (linear in inputs, not in weights as in NTK), and a proxy deviation bound. The bound of [3] is good when the width is large (hence the finite-width NTK is close to the limit NTK), while our bound is good when the activation functions are close to be linear (hence the proxy does not deviate from the original model much).
> >
> > We will include a short discussion of differences between NTK (linear in weights) and linear networks (linear in inputs) into the paper to avoid further confusion among readers.
> >
> > [3]: Arora et al., 2019: Fine-Grained Analysis of Optimization and Generalization for Overparameterized Two-Layer Neural Networks http://proceedings.mlr.press/v97/arora19a/arora19a.pdf

---

> ### Author Response · Authors · 2024-09-25
> **Revision: increased figure sizes**
>
> Dear reviewer,
>
> Following your request, we have increased figure sizes (this way, the font sizes in the figures also became larger, compared to the main text font).

---

### Review · Reviewer_5zEF · 2024-09-12

**Summary Of Contributions:**

The authors aim to present generalization bounds that become meaningful for networks that approximate linearity. This is particularly valuable as it does not require the actual training of the model to evaluate these bounds, making them a priori—an approach not previously used for neural networks to this extent.

The paper emphasizes the difficulty in understanding why neural networks generalize well on unseen data despite their massive capacity and complex training dynamics. It aims to address the shortcoming of classical methods that uniformly bound performance over model classes, which often prove inadequate for networks with poor parameter configurations.

**Audience:**

Yes

**Claims And Evidence:**

Yes

**Requested Changes:**

Comparisons with some key previous work are missing.

1. Such as neural tangenet kernel, which is also a linear surrogate of neural networks for analysis purpose. "Jacot, A., Gabriel, F., & Hongler, C. (2018). Neural tangent kernel: Convergence and generalization in neural networks. Advances in neural information processing systems, 31."

2. Ridgeless regression can also interpret the generalization gap of over-parameterized networks (under certain assumption, this result could also extend to DNNs). Liang, Tengyuan, and Alexander Rakhlin. "Just interpolate: Kernel “Ridgeless” regression can generalize." The Annals of Statistics 48, no. 3 (2020).

Could you please compare your results against them.

**Strengths And Weaknesses:**

Some advantages we can observe:
1. The bound does not depend on a held-out dataset, which is a novelty among existing generalization bounds.
2. It does not scale with network width, unlike some other bounds that exhibit such growth.
3. It focuses on the generalization gap of the original trained model rather than relying on proxies or external alterations to the model weights.

Some limitations:
1. The bound becomes useful only under specific conditions, such as when the functions are close to linear, and when a simple linear model's generalization bound is non-vacuous in similar settings.
2. The application might be limited as these conditions are not always met, such as for binary classification tasks on high-dimensional data like MNIST without modification.

The paper clearly outlines that certain assumptions, such as whitening of data, make the theoretical analysis cleaner and simpler. However, it acknowledges that this assumption might not always hold in practical scenarios.

The authors compare their approach to existing literature, highlighting how their improvements offer practical benefits like eliminating the need for a trained model to compute bounds and not being affected by network width. In some degree, NTK can be also viewed as a type of linear neural network (or linear surrogate of neural network). Have you compared your results with NTK?

---

> ### Author Response · Authors · 2024-09-19
> **Response to Reviewer 5zEF**
>
> We thank the anonymous reviewer for valuable comments!
>
> Concerning the suggested comparison with NTK, we believe there is a terminological confusion. Our proxy models use weights of a trained linear network, where "linear" means "without activation functions". That is, it is linear in its inputs, $x$, but not its weights. Whereas in the NTK limit, the model is trained as if it was linear in weights, $\theta$. Training of the model becomes equivalent to the training of a kernel method with NTK being a kernel. This kernel is not linear, hence the trained model is not linear in $x$.
>
> We emphasize that the proxy models we use, besides using weights of a trained linear model, are not linear themselves.
>
> We also emphasize that for the network to be close to its NTK limit, it should be either (a) very wide and parameterized in a certain (non-standard) way, or (b) initialized with very large weights (also not following the standard practice). Otherwise, there is still an associated kernel (also called NTK, but not to be confused with "limit NTK"), but the kernel is stochastic (since weights are initialized randomly) and evolves throughout training. Moreover, the evolution of NTK (not present in the NTK limit) is favorable as it is associated to feature learning.
> In our approach, we require neither special parameterization nor large width, while the associated kernel does evolve since the model is not linear in weights.
>
> Therefore our approach is quite orthogonal to NTK and generalizability of kernel methods. In terms of high-level methodology, the closest paper we could mention is [1], where they consider a two-layer NTK-parameterized MLP, and prove a generalization bound for it when width is sufficiently large. For this, they combine a generalization bound for the infinite-width limit when the kernel is constant, and a term that takes into account the fact that the kernel for finite-width deviates from the limit one. Whereas what we do is we combine a generalization bound for a proxy model that exploits the weights of a linear network (linear in inputs, not in weights as in NTK), and a proxy deviation bound. The bound of [1] is good when the width is large (hence the finite-width NTK is close to the limit NTK), while our bound is good when the activation functions are close to be linear (hence the proxy does not deviate from the original model much).
>
> We will include a short discussion of differences between NTK (linear in weights) and linear networks (linear in inputs) into the paper to avoid further confusion among readers.
>
> [1]: Arora et al., 2019: Fine-Grained Analysis of Optimization and Generalization for
> Overparameterized Two-Layer Neural Networks http://proceedings.mlr.press/v97/arora19a/arora19a.pdf

---

### Review · Reviewer_tWuH · 2024-09-16

**Summary Of Contributions:**

The paper studies generalization error for multi-layer fully-connected neural networks by studying its approximation by linear neural networks. Upper bounds are derived for the gap between training and test error, for multi-layer neural networks trained by gradient flowm, within a fixed total training time. The bound contains two terms, corresponding to the generalization of a linear neural network, and the approximation error between NNs with linear and nonlinear activation functions.

The generalization error of neural networks is a central question in machine learning theory. However, the results derived in the paper are relatively weak. In particular, the last term in the generalization bounds in Theorem 4.2 grow very quickly as the training epoch and model size increases, effectively making it vacuous. With a reasonably long training time for the neural networks, it is not clear why the proposed bounds could be better than usual bounds derived using VC dimensions and Rademacher complexities. Moreover, the method of linear approximation for proving generalization bounds is fundamentally limited (see weakness below). Therefore, I don't think this paper could be publishable at TMLR.

**Audience:**

No

**Broader Impact Concerns:**

No ethical concerns are involved.

**Claims And Evidence:**

Yes

**Requested Changes:**

More comparison with existing literature might help. Also, the authors need to explain why we don't use a linear model directly since they bound the difference between NN and a linear model.

**Strengths And Weaknesses:**

Strength. The generalization bounds derived in the paper are not seen in existing literature.

Weakness. The last term in Theorem 4.2 exhibit several aspects of weakness.
- The bound grows exponentially with number of layers. Note that using standard tools of Rademacher complexities and norm-based bounds, we can already derive an exponentially-growing bound. But the norm-based bound does not suffer from other weakness and does not require early stopping.
- The bound grows exponentially with time (for 2 layers) and blows up within finite time (for more than 3 layers). This requires the training time to be very short. Within such as short training time, the neural network can still be far from convergence to a local minimum.
- This term does not decrease with sample size.
Moreover, the paper establishes the generalization bound based on approximating a non-linear NN with a linear NN. This approach does not seem to be able to yield good bounds. In particular, if there is a regime where the bounds derived in this paper are small, that means the nonlinear NN is well-approximated by a linear model, and the approximation error appears in the final excess risk bound. In such a case, why don't we simply use a linear model? (The linear model may approximate the true model worse than NN, but here we are paying for approximation error anyway).

---

> ### Author Response · Authors · 2024-09-23
> **Repsonse to Reviewer tWuH**
>
> Thank you for your comments!
> We would like to comment on the weakness points you mentioned:
>
> 1. **The bound grows exponentially with number of layers. Note that using standard tools of Rademacher complexities and norm-based bounds, we can already derive an exponentially-growing bound. But the norm-based bound does not suffer from other weakness and does not require early stopping.**
>
> We agree with this statement. However, Rademacher complexity of a class of models realizable by a given neural network grows with width, while our bound does not. For this reason, while Rademacher complexity of this class could be indeed computed a-priori, it is numerically huge for all neural nets of reasonable size, thus making the bound vacuous. One could indeed restrict the model class, e.g. to neural nets with weight norms not exceeding a given value as in [1,2], thus reducing the complexity, however, (1) the resulting bound is still vacuous, and (2) it is not an a-priori bound anymore.
>
>
> 2. **The bound grows exponentially with time (for 2 layers) and blows up within finite time (for more than 3 layers). This requires the training time to be very short. Within such as short training time, the neural network can still be far from convergence to a local minimum.**
>
> This is totally true: our bound indeed grows with training time. Nevertheless, it could be non-vacuous at the beginning of training: in Section 4.3 we demonstrate that our bound stays non-vacuous up to $t \leq t_\alpha^*(\beta)$, the linear network training time, and small $\epsilon$.
>
>
> 3. **This term does not decrease with sample size. Moreover, the paper establishes the generalization bound based on approximating a non-linear NN with a linear NN. This approach does not seem to be able to yield good bounds. In particular, if there is a regime where the bounds derived in this paper are small, that means the nonlinear NN is well-approximated by a linear model, and the approximation error appears in the final excess risk bound. In such a case, why don't we simply use a linear model? (The linear model may approximate the true model worse than NN, but here we are paying for approximation error anyway).**
>
> We would like to emphasize that the proxy models we use to prove our bounds are not linear, even though they use weights of a trained linear model, $\theta^0$. We could indeed use the trained linear model, $f^0_{\theta^0}$, as a proxy too. Then the deviation term, $\Delta$, would be $O(\epsilon)$, same as for $f^\epsilon_{\theta^0}$ which corresponds to $\kappa=1$. However, the constant of this $O(\epsilon)$ will be larger; also, note that the deviation is of the order of $O(\epsilon^2)$ for the $\kappa=2$ proxy model which is guaranteed to be smaller for small enough epsilon. In our experiments we see that $\kappa=1$ gives a larger bound than $\kappa=2$; therefore we do not expect the linear model, $f^0_{\theta^0}$, to result in a better bound than the proxy for $\kappa=2$. Our experiments, see Figure 8, top right, confirm this intuition.
>
> If we still decided to use a linear model as a proxy, we could probably bound the generalization gap of the proxy, $\Upsilon$, better than using a naive counting-based bound as we did. However, bounds for linear regression we are aware of require strong assumptions on the data distribution (e.g. sub-Gaussian features and noisy linear teacher that generates $y$), see e.g. [3].
>
>
> [1] Bartlett P., Foster D.J., and Telgarsky M., 2017: Spectrally-normalized margin bounds for neural networks https://papers.nips.cc/paper_files/paper/2017/hash/b22b257ad0519d4500539da3c8bcf4dd-Abstract.html
>
> [2] Neyshabur, B., Bhojanapalli, S., and Srebro, N., 2018: A PAC-bayesian approach to spectrally-normalized margin bounds for neural networks https://openreview.net/forum?id=Skz_WfbCZ
>
> [3] Tsigler and Bartlett, 2023: Benign overfitting in ridge regression https://www.jmlr.org/papers/v24/22-1398.html

---

### Author Response · Authors · 2024-09-24
**Revision: extended literature review**

Dear reviewers,

As some of you requested, we have extended the literature review.

To be specific, we have included a section about existing a-priori and a-posteriori bounds, and a section that emphasizes the fundamental differences between our approach based on linear networks (linear in inputs), and NTK-based approaches (linearized training procedure, linear in weights).

---

### Decision · Action_Editor_bb4P · 2024-12-11

**Recommendation:** Accept as is

**Comment:**

This paper provides an a-priori generalization bound for nearly-linear neural networks. The bound is shown to be non-vacuous under certain conditions, and is investigated empirically. Generally, the reviewers did not raise issue with the correctness or rigor of the results, but were divided about whether the approach and the findings were useful and interesting. The topic receiving the most discussion was the focus on near-linearity and the reliance on linear models, and whether this focus rendered the results uninteresting in the context of previously-established related bounds. The authors replied that this paper provides a first non-vacuous a-priori bound useful for neural networks, and that other prior bounds cannot exploit near-linearity. This rationale seems to solidify the current contribution as a useful and interesting one, and as such I recommend acceptance.

**Audience:**

Yes, some individuals would find this paper interesting

**Claims And Evidence:**

Yes, the claims are well-supported by the evidence